# Effects of mild-to-moderate hearing loss and linear amplification on vocal emotion recognition in middle-aged-older individuals

**Mattias Ekberg**[1]*, **Josefine Andin**[1], **Carine Signoret**[1], **Stefan Stenfelt**[2], **Örjan Dahlström**[1]

**1** Department of Behavioural Sciences and Learning, Linköping University, Linköping, Sweden,
**2** Department of Biomedical and Clinical Sciences, Linköping University, Linköping, Sweden

\* henrik.mattias.ekberg@liu.se

## Abstract

Previous research has shown deficits in vocal emotion recognition in sub-populations of individuals with hearing loss. As emotion recognition is an essential ability that affects social interaction, and in extension, can impact well-being, understanding vocal emotion recognition difficulties is a high priority research topic. Furthermore, it has been shown that although hearing aids improves word recognition, it does not improve emotion recognition. To explore emotion recognition and the effect of amplification in individuals with hearing loss, we examined recognition of vocal emotions expressed both verbally and non-verbally in middle-aged to older individuals with and without linear amplification (similar amplification across sound levels). Twenty-one experienced hearing aid users with bilateral mild-to-moderate hearing loss, and 20 individuals with normal hearing performed a vocal emotion recognition task on sentences and non-verbal vocalizations. The hearing loss group had poorer emotion recognition for both sentences ($F(1,38)=15.24$, $p < 0.001$, $\eta^2_p=0.29$, without linear amplification, and $F(1, 38)= 5.62$, $p=0.023$, $\eta^2_p=0.13$, with linear amplification) and non-verbal vocalizations($F(1,38)= 25.18$, $p < 0.001$, $\eta^2_p=0.40$, without linear amplification, and $F(1, 38)= 10.30$, $p=0.003$, $\eta^2_p=0.21$, with linear amplification). However, linear amplification significantly improved the recognition of happiness ($p < 0.001$), which is distinguished by frequency parameters, for sentences. For non-verbal vocalizations, recognition of fear ($p = 0.004$) and anger ($p = 0.004$), were improved by linear amplification. Patterns of confusion were similar for the two groups, which may suggest that both groups perceived the emotions similarly, but that the degree of perceptual precision was lower in the hearing loss group. In sum, hearing loss negatively impacts vocal emotion recognition, but linear amplification can enhance recognition for some emotions.

## 1. Introduction

Emotion recognition in individuals with hearing loss has been identified as a high-priority research topic [1]. This topic includes, for example, questions about how vocal emotion

**Data availability statement:** All data underlying the findings described are fully available on the Open Science Framework OFC); https://osf.io/,DOI10.17605/OSF.IO/V3ANK. Project name: "Vocal emotion recognition in middle-aged-to-older individuals with sensorineural hearing loss and normal hearing individuals"

**Funding:** Örjan Dahlström was partly funded through a grant from The Swedish Association of Hard of Hearing People. Ref 2016-531. https://hrf.se/ The Swedish Association of Hard of Hearing People did not play any role in the design, data collection, analysis, preparation or decision to publish the manuscript.

**Competing interests:** The authors have declared that no competing interests exist.

recognition deficits associated with hearing loss may affect well-being, the relationship between vocal emotion recognition in individuals with hearing loss and psychoacoustics, and the impact of hearing technologies such as hearing aids, and auditory rehabilitation [1]. In the present study, we examine the effects of mild-to-moderate hearing loss and linear amplification on vocal emotion recognition for emotions expressed verbally in spoken sentences and non-verbally in spoken expression (non-verbal vocalizations) in middle-aged-to-older individuals and how this is related to the acoustic characteristics of the emotions.

## 1.1. Vocal emotion recognition

Emotions are defined as "episodic, relatively short-term, biologically-based patterns of perception, experience, physiology, action and communication that occur in response to specific physical and social challenges and opportunities" [2, p. 468]. Emotions can be communicated in several ways, for example through vocalizations, i.e., vocal emotion expressions. The recognition of vocal emotion expressions depends on an interplay between different factors [3]. The person expressing an emotion produces different lower-level features (distal cues), which are registered by the listener and integrated into higher-level perceptual features (proximal cues). The higher-level features are then interpreted by the listener according to implicit rules (cognitive processing). The rules for interpreting high-level features are influenced by the individual's knowledge and sociocultural context [3]. Furthermore, individuals' psychobiological architecture, including for example the auditory system's functioning, also affects vocal emotion expression and perception [3]. Vocal emotion expressions can be divided into verbal prosody and non-verbal vocalizations [4,5]. Prosody consists of modulations of acoustic characteristics such as frequency, amplitude, and duration that occur over longer timescales than the words in speech (suprasegmentally) [4,6], while non-verbal vocalizations consist of brief, non-linguistic sounds such as laughter, expressing happiness [4]. The expression of emotional prosody is likely affected to a high degree by socioculturally grounded norms of communication, while nonverbal expressions are mainly shaped by involuntary physiological changes [3].

Non-verbal vocalizations are generally recognized with higher accuracy than emotional prosody [3,7], but accuracy for different emotions varies for both emotional prosody [8–11] and non-verbal vocalizations [12–16]. The emotional prosody of anger, fear, sadness, and happiness have consistently been found to be recognized with above-chance accuracy and are seldom confused with other emotions [3,8–11]. Recognition of emotions in speech can also be supported by speech understanding, such that recognition is higher when the meaning of speech, in addition to emotional prosody, indicates the speaker's intended emotion expression [17]. The ability to recognize others' vocal emotion expressions play an important role in social interaction, and difficulties in vocal emotion recognition can therefore influence individuals' participation and subjective well-being [1].

## 1.2. Acoustic characteristics of vocal emotions

Different emotions have different acoustic characteristics, and vocal emotion recognition depends on the listeners' ability to perceive the acoustic features of emotions [3]. In speech, anger, happiness, fear, and surprise have, for example, been described as having high mean amplitude and relatively high pitch [10,11,18–20], while sadness has been described as having low pitch and low amplitude [10,19]. For non-verbal vocalizations, anger has been shown to have relatively low mean pitch, compared to several other emotions, while fear has low pitch variation and sadness has slightly lower mean intensity compared to several other emotions

[12,16]. Some studies have also examined which acoustic parameters contribute to accurately recognizing emotions [e.g., 9,12,16]. For example, it was shown that accurate recognition of anger, happiness and fear in speech was positively associated with pitch (logarithmic F0) and loudness (estimated intensity) [10]. Another example is that recognition of the non-verbal vocalization of anger was found to be negatively associated with pitch, while recognition of fear was positively associated with pitch [12]. Examining which acoustic characteristics distinguish different emotion categories may provide further insights into vocal emotion recognition.

### 1.3. Prevalence and causes of hearing loss

Hearing loss is among the leading causes of disability globally [20,21]. It leads to difficulties in communication, which can contribute to social isolation and reduced quality of life [1]. It has been estimated that 18% of the population in Sweden have a hearing loss [22]. The prevalence of hearing loss increases with age and is around 24% in middle-aged-to-older individuals (ages 55–74) [22].

Hearing loss is commonly determined by the pure-tone average, which is measured across several frequencies) [23]. The specific frequencies included in measuring PTA varies, but it is common to include the frequencies of 500, 1000, 2000, and 4000 Hz, which is referred to as PTA-4 [23]. According to one description, degrees of hearing loss a PTA between 26–40 dB HL constitutes a mild hearing loss, between 41–60 dB HL a moderate hearing loss, between 61–80 dB HL a severe hearing loss, and >81 dB HL a profound hearing loss) [21]. The majority of individuals with hearing loss have mild-to-moderate hearing loss [21].

Hearing loss can occur by any damage to the auditory system [24]. The most common cause of hearing loss is peripheral sensorineural hearing loss, which refers to damages to the cochlea and the auditory nerve [25–27]. This commonly occurs due to the negative effects of aging on the auditory system, which is referred to as age-related hearing loss (presbycusis; [20]). Age-related hearing loss is particularly associated with elevated high-frequency hearing thresholds of 2000 Hz or higher [24].

### 1.4. Perceptual effects of peripheral hearing loss

Hearing loss leads to reduced hearing sensitivity [28]. Peripheral sensorineural hearing loss further leads to loss of dynamic range, reduced frequency selectivity, and poorer pitch perception [25,26]. The reduced hearing sensitivity makes sounds become less audible in specific frequency bands. A loss of dynamic range means the quiet sounds may be inaudible while loud sounds are perceived similarly as by normal hearing individuals. Reduced frequency selectivity makes it difficult to distinguish between sounds of different frequencies [25–26].

### 1.5. Hearing aids

The use of hearing aids is the most common intervention for individuals with mild-to-moderate hearing loss [21,29]. It improves their listening ability and verbal communication [21,29]. Despite its benefits, hearing aids does not normalize auditory perception [25,28]. Linear amplification (used mostly in older hearing aids) involves amplification of all sounds regardless of level. This restores the audibility of quiet sounds, however, loud sounds may become too loud [25]. Modern hearing aids use non-linear amplification and compression. This involves the selective amplification of quieter sounds, which resolves the problem of sounds becoming too loud [28]. However, evidence does not support hearing aids, including modern hearing aids, restoring frequency selectivity and pitch perception [25].

## 1.6. The impact of hearing loss on vocal emotion recognition

Vocal emotion recognition is reduced in individuals with hearing loss in general [30–34] including in mild-to-moderate hearing loss [31–33]. For example, poorer recognition for individuals with mild-to-severe hearing loss compared to individuals with normal hearing have been found in young and middle-aged individuals (ages 22–74; [31]) and in middle-aged to older adults (50–90 years old; [31,33].

Further, Goy et al. found that individuals with mild-to-moderate hearing had better recognition of sadness compared to anger, fear, and happiness, but not surprise [31].

While the studies presented here show that individuals with hearing loss, including mild-to-moderate hearing loss, have poorer vocal emotion recognition, questions remain regarding its causes. For example, low frequency hearing loss contributed only marginally to emotion prosody recognition in individuals with mild-to-moderate hearing loss, while high frequency thresholds did not contribute at all [30]. In addition, Goy et al. did not find a significant association between the PTA and emotion recognition in individuals with mild-to-moderate hearing loss [31]. However, Legris et al., examining recognition of non-verbal vocalizations, did find a significant negative association between PTA and recognition accuracy in this group [33]. As such, the association between hearing sensitivity and vocal emotion recognition in individuals with hearing loss might differ between emotions in speech and nonverbal expressions. Further, the evidence so far shows that the use of hearing aids does not normalize vocal emotion expressions, neither in speech [31,35], nor non-verbal vocalizations [33]. Goy et al [31] found small, non-significant, improvements in the recognition of some emotions, particularly fear, [31], and Legris et al. found overall improvements but with remaining significantly poorer recognition compared to normal hearing individuals [33]. It is unclear how important different factors, such as reduced hearing sensitivity, deficits in psychoacoustic abilities, and changes in cognitive processing are for explaining vocal emotion recognition deficits in individuals with mild-to-moderate hearing [32].

Several aspects of hearing aid signal processing may influence vocal emotion recognition. In the current study, the audibility of the sound provided by hearing aid amplification is investigated. This is achieved through linear amplification based on individuals' hearing thresholds using the Cambridge formula [36]. The Cambridge formula aims to restore loudness perception across the entire speech spectrum for individuals with hearing impairment. An advantage of the Cambridge formula compared to other formulae for linear amplification (e.g., NAL-R) is that it provides slightly larger gains for higher frequencies (>2000 Hz; [37]), which could be particularly beneficial for individuals with age-related hearing loss.

Furthermore, the ability to recognize emotions is related to the processing of the acoustic characteristics of different emotion categories, therefore relating recognition as well as how different specific emotions are mistaken for other emotions in individuals with mild-to-moderate hearing loss to the acoustic characteristics of different emotions may provide further insight into vocal emotion recognition in this group.

## 1.7. Aims

In this study, we examine recognition accuracy for different emotions (i.e., which emotions are the easiest or most difficult to accurately recognize) and patterns of confusion (i.e., which emotions are mixed up when misrecognized) for individuals with normal hearing and for individuals with hearing loss (listening with and without linear amplification), and relate performance to acoustic analyses, the aims are to gain a deeper understanding of;

- how the effect of hearing loss affects emotion recognition (by comparing performance of participants with and without hearing loss), and;

- how emotion recognition is affected by linear amplification (by examining performance of participants with hearing loss using linear amplification).

The results regarding performance will be related to acoustics in order to examine whether vocal emotion acoustics provide clues to better understand the effects of hearing loss and amplification on the recognition of different emotions, in relation to known perceptual consequences of sensorineural hearing loss.

In the stage 1 registered report protocol for this article [38], we made several predictions based on the research that was available to us at that time. In line with general guidelines for registered reports and open science practices, we have not changed our predictions, and this article focuses on examining the predictions we originally made.

Based on the literature, we predicted that:

1. Individuals with hearing loss will have poorer recognition compared to normal hearing individuals for emotions expressed verbally, regardless of acoustic features and regardless of the use of linear amplification, and for non-verbal vocalizations when linear amplification is not used

2. Individuals with and without hearing loss will not differ in accuracy for non-verbal vocalizations when linear amplification is used (This prediction is contrary to the findings of Legris et al [33] of which we were not aware at the time the predictions were formulated (Stage 1 of the registered report)..

3. Vocal emotions, which are more distinct in terms of acoustic parameter measures, will be recognized with higher accuracy for both groups, but emotions that are distinguished mainly based on frequency parameters will be less accurately recognized by the hearing loss group.

4. The more distinct the frequency-related acoustic parameters of an emotion are, the better that emotion will be recognized when linear amplification is used compared to when not.

5. Patterns of confusion will differ between individuals with and without hearing loss for both verbally and non-verbally expressed emotions.

## 2. Methods

### 2.1. Participants

Using a 2 x 6 mixed design (group x emotion), 80% power, 5% significance level, correlation between repeated measures of 0.5 and with N=56 participants (a reasonable number of participants from a recruitment point of view), we would be able to detect an effect as small as $\eta^2$=.02 (f=0.14); a next to small effect size. Power analysis was performed using G*Power version 3.1.9.7 [39].

Twenty-six native Swedish-speaking participants, aged 40–75, with mild-to-moderate, bilateral, symmetric sensorineural hearing loss (PTA4, based on 0.5, 1, 2, and 4k Hz, of 30–60 dB HL), who have been using hearing aids for at least one year, and 34 age-matched native Swedish speaking participants with self-reported normal hearing were recruited. We originally intended to recruit at least 28 individuals from both groups. Approximately equally many men and women were included in both groups. The audiometric profiles of participants with hearing loss and information about their hearing aids, were planned to be obtained through the audiological clinics, from participants who gave their consent. However, this was not practically solved, so instead audiometric profiles of all participants were obtained through air-conduction pure-tone audiometry by the first author. Since an association between general cognitive ability (G) and emotion recognition ability have been established [40], the subtest Matrices from the Swedish version of WAIS-IV, which is strongly correlated with G, was used for all participants [41]. Before being invited to participate in the study, interested individuals

filled out an online questionnaire, including questions about health problems, hearing-aid use, different diagnoses, educational attainment, age, gender, and native language. The survey contained questions regarding whether individuals suffered from hyperacusis, neurological disorders affecting the brain (e.g., multiple sclerosis or epilepsy), severe tinnitus (which is perceived to cause impairment and disability), developmental psychiatric disorders (e.g., ADHD, autism spectrum disorders or intellectual disability), mood and anxiety disorders (e.g., social anxiety disorder or depression), the experience of great difficulties in identifying and describing one's own emotions. In addition, individuals with hearing loss also responded to the question of whether conductive problems (exemplified by ear infections, damage to the ear drum, fluid in the ear, and otosclerosis) were the cause of their hearing loss.

**2.1.1. *Inclusion criteria.*** Inclusion criteria for the hearing loss group were a PTA4 between 30 dB–60 dB HL for both ears), and hearing aid use experience (having used a hearing aid for at least one year). Inclusion criterion for the normal hearing group was normal hearing, indicated by hearing thresholds of < 20 dB HL at all frequencies between 125 and 4000 Hz, and no worse than 30 dB HL at 8000 Hz. For both groups, participants had to be between 40 and 75 years old. Initially the age criterion was 50–75, but this was changed in order to include more participants with normal hearing.

**2.1.2. *Exclusion criteria.*** Exclusion criteria for both groups included low general intelligence as indicated by performance below two standard deviations from the mean for their age span on the subtest Matrices and self-reported neurological, developmental and neuropsychiatric diagnoses (see information about the survey above). Further, individuals who self-reported that conductive issues were the cause of their hearing loss and individuals with asymmetric hearing loss (a difference between the two ears in PTA-4 of ≥15 dB) were excluded. Other reasons for exclusion were hyperacusis, neurological disorders affecting the brain (e.g., multiple sclerosis or epilepsy), severe tinnitus (which is perceived to cause impairment and disability), developmental psychiatric disorders (e.g., ADHD, autism spectrum disorders or intellectual disability), mood and anxiety disorders (e.g., social anxiety disorder or depression), or the experience of great difficulties in identifying and describing one's own emotions.

Fourteen participants who had reported normal hearing and five participants with hearing loss did not fulfill the criteria and were excluded. Thus, results from a total of 41 participants (21 with hearing loss, and 20 with normal hearing) were included in the study. Recruitment of participants began on October the 10th 2023 and ended February the 5th 2024.

All participants received both written and spoken information about the study before giving consent. Participants gave written consent through signing a letter of informed consent. We followed the declaration of Helsinki, and the project is approved by the Swedish Ethical Review Authority (Dnr: 2020–03674).

## 2.2. Material and procedure

**2.2.1. *Stimuli material.*** The stimulus material is based on fourteen emotionally neutral sentences from the Swedish version of the hearing in-noise test (HINT; [42]), and non-verbal vocalizations. Four actors – an older female (69 years old), an older male (73 years old), a young female (19 years old) and a young male (29 years old) – were recorded when reading the sentences and when producing non-verbal expressions (sound expressions without using language), all with emotional prosody expressing different emotions of high and low intensity. The emotions included in the recordings are anger, happiness, sadness, fear, surprise, and interest. For the sentences, prosodically neutral versions were also recorded. For the non-verbal vocalizations, the actors were instructed to imagine themselves experiencing different emotions and to make expressions with sounds, without language, which match those emotions. This entails some variation of the specific sounds expressed for specific emotions by different

participants. The stimuli were recorded in a studio at the Audiology clinic at Linköping University Hospital, with the aid of a sound technician. Recordings were made in Audacity™ [43] using high-quality equipment, 24-bit resolution, and a 44.1 kHz sampling rate. From each actor, the clearest and cleanest recordings out of two or more repetitions for each sentence and non-verbal expression was selected. With very few exceptions the sentences are approximately 2–3 seconds long and non-verbal expressions are 1–2 seconds long. The stimuli were equalized in terms of audibility. The calibration (equalization) was based on performance by 10 normal hearing participants on a recognition test presenting the stimuli at different signal-to-noise-ratios (SNR). Based on an online administered vocal emotion recognition task [44], sentences with at least 50% accurately recognized emotions were used and therefore interest was excluded for sentences and surprise was excluded for non-verbal vocalizations in the present study. The emotion categories in the present study are thus anger, happiness, fear, sadness, and surprise for sentences, and anger, happiness, fear, sadness, and interest for non-verbal vocalizations.

**2.2.2. Procedure.** Stimuli were presented bilaterally at a level of 65 dB SPL. This level was consistent for NH participants and HL participants in the unaided listening condition and served as the baseline level in the aided listening condition. The stimuli were delivered through AKG K182 headphones, connected to a Supelux HA3D headphone amplifier. In the aided listening conditions, the audio files were preprocessed in MATLAB v. R2022a using an algorithm that implemented the Cambridge formula for linear hearing aids [36], based on each participant's audiogram. These filtered audio files were then presented to individuals with hearing impairment, simulating the experience of hearing through hearing aids with the specific gain function. Participants sat in front of a computer screen and were presented with the written question "Which emotion was expressed in the recording you just heard?" with the options; happiness, anger, fear, sadness, and surprise for sentences, and happiness, anger, fear, sadness, and interest for non-verbal expressions. The task was to select the correct emotion by button-press on a keyboard. Two seconds after the participant's response, the next trial was presented. The purpose of the lag between response and stimulus presentation was to allow for shifting of attention from responding to listening. The task did not proceed unless participants responded, in order to avoid missing data. Participants could, however, choose to abort the task. Stimuli were presented using PsychoPy version 3.0.

Sentences and non-verbal vocalizations were divided into separate sessions, and participants had the opportunity to pause briefly between sessions. Participants with hearing loss performed a total of four sessions; two sessions with linear amplification (aided listening condition), one for sentences and one for non-verbal expressions, and two sessions without linear amplification (unaided listening condition), one for sentences and one for non-verbal expressions. The order of listening conditions was balanced across participants. To have the same test-time and load, participants with normal hearing also performed all sessions, two with sentences as stimuli and two with non-verbal vocalizations. However, participants with normal hearing performed two of the sessions, one with each stimulus type, with noise-vocoded stimuli instead of linear amplification. The results from the vocoded stimuli are not included in analyses. Stimuli within sessions were presented in randomized order. Two different sets of 54 sentences each, and two different sequences of the 20 non-verbal expressions were used for comparison within each listening condition. Thus, there were in total 108 sentences (24 with anger, 25 with happiness, 22 with fear, 21 with sadness, and 16 with surprise) and 20 non-verbal vocalizations (4 with anger, 4 with happiness, 4 with fear, 4 with sadness, and 4 with interest).

## 2.3. Analyses

**2.3.1. Acoustic analyses.** For each recording of verbal and non-verbal emotions, the acoustic parameters of the Geneva Minimalistic Parameter Set (GeMAPS, [45]) were extracted

using the OpenSmile toolkit v.2.3 [46]. The GeMAPS consists of a set of acoustic parameters that have been proposed as a standard for different areas of automatic voice analysis, including the analysis of vocal emotions. The parameters of GeMAPS have been shown to be of value for analyzing emotions in speech in previous research [45] and are described in Table 1.

To control for baseline parameter differences between speakers' voices, z-scores were calculated for each parameter and for each sentence, using the means and standard deviations of each speaker's neutral voice [8]. For non-verbal vocalizations, the same procedure was followed, with the exception that the means and standard deviations of non-verbal vocalizations were used as reference.

**2.3.2. *Statistical analyses.*** First, comparisons between the normal hearing group and the hearing loss group listening without amplification in vocal emotion recognition were made using one 2 x 5 mixed ANOVA, for sentences, and one 2 x 5 mixed ANOVA, for non-verbal vocalizations, all with Group (HL vs NH) as the between-subjects variable and Emotion as the within-subjects variable (anger, happiness, fear, sadness and surprise for sentences; anger, happiness, fear, sadness, and interest for nonverbal expressions). The same analyses were then repeated but using sessions with linear amplification for the hearing loss group. These four analyses included age as a covariate (see 2.1 Participants).

Table 1. Description of the acoustic parameters of the GeMAPS as discussed in Eyben et al. [45].

| Parameters | Explanation |
|---|---|
| *Frequency related* | |
| Pitch (PT) | the logarithmic fundamental frequency, $F_0$, on a semitone scale starting at 27.5 Hz |
| Jitter | deviations in individual consecutive $F_0$ period lengths |
| Frequency – formant 1 | the center frequency of the first formant |
| Frequency – formant 2 | the center frequency of the second formant |
| Frequency – formant 3 | the center frequency of the third formant |
| Bandwidth – formant 1 | The bandwidth of the first formant |
| *Energy/Amplitude/Intensity related* | |
| Shimmer | difference of the peak amplitudes of consecutive $F_0$ periods |
| Loudness | an estimate of the perceived signal intensity from an auditory spectrum |
| Harmonics-to-noise ratio | relation of energy in harmonic components to energy in noise-like components |
| *Spectral (balance)-related components* | |
| Alpha ratio | ratio of the summed energy from 50–1000 Hz and 1–5 kHz |
| Hammarberg index | ratio of the strongest energy peak in the 0–2 kHz region to the strongest energy peak in the 2–5 kHz region |
| Spectral slope 0–500 Hz | linear regression slope of the logarithmic power spectrum within the given band |
| Spectral slope 500–1500 Hz | linear regression slope of the logarithmic power spectrum within the given band |
| Relative energy – formant 1 | the relative energy of the first formant and the ratio of the energy of the spectral harmonic peak at the first formant's center frequency to the energy of the spectral peak at the fundamental frequency |
| Relative energy – formant 2 | The relative energy of the second formant and the ratio of the energy of the spectral harmonic peak at the second formant's center frequency to the energy of the spectral peak at the fundamental frequency |
| Relative energy – formant 3 | The relative energy of the third formant and the ratio of the energy of the spectral harmonic peak at the third formant's center frequency to the energy of the spectral peak at the fundamental frequency |

Second, the effects of linear amplification on recognition were analyzed within the hearing loss group by a 2 x 5 repeated measures ANOVA for verbal stimuli, and a 2 x 5 repeated measures ANOVA for non-verbal vocalizations, both with listening condition (amplified vs. non-amplified) and emotion as factors.

Significant effects were followed by Šidák post hoc-test. In the case of a violation of the assumption of sphericity, Huynh-Feldt corrected degrees of freedom and corresponding $p$-values were reported.

The outcome variable in all analyses were raw accuracy proportions (ratios of accurate responses for each emotion), referred to as "emotion recognition". However, in addition, we also report Rosenthal's Proportion Index (Rosenthal's PI; [47]) for each emotion expression of both stimulus types. Rosenthal's PI is a measurement of the accuracy rate for response data which spans between 0.5 and 1 and is calculated based on the obtained rate in relation to the number of response options. A specific accuracy rate for a response category will be calculated as higher when there are more response options available and lower if there are fewer response options [47]. This enables easier comparison between studies with different number of response options. In addition to raw accuracy proportions, we also present confusion matrices that represent the distribution of all responses in percentages.

For emotions expressed in sentences we also analyzed which acoustic features significantly contributed to discriminating emotions from one another. This was done in two steps. First, Principal Component Analyses (PCA) with VARIMAX rotations were performed for the parameters in each acoustic domain separately (keeping as many components as there were eigenvalues > 1, excluding components with only one variable). Second, the distinctiveness of pairwise emotions were described by the differences between emotions' mean values of components, and the Euclidean distance between emotions when using the mean values as coordinates. Finally, to examine the predictive ability of the acoustic features, the components were used in multinomial regression models with emotion type as the dependent variable (analyses were repeated with each emotion type as a reference).

Separate confusion matrices were calculated for each combination of group, stimulus type, and listening condition (amplification or no amplification).

The statistical analyses were performed in IBM SPSS v. 29.0.2.0. The material and scripts used in the present study are openly available on the Open Science framework (OSF) at https://osf.io/, Identifier: DOI 10.17605/OSF.IO/V3ANK. Project name: "Vocal emotion recognition in middle-aged-to-older individuals with sensorineural hearing loss and normal hearing individuals".

## 3. Results

### 3.1. Demographic data

Demographic data and hearing thresholds of the participants are presented in Table 2. The mean age of the hearing loss group (M=63, SD=8) was higher than for the normal hearing group (M=52, SD=8), $t(39)$=4.42, $p$<.001, $d$=1.38. Therefore, to control for the potential influence of age on group differences, age was included in all statistical analyses including group as a factor. The distribution of gender $\chi2$=0.51, $p$=.48, and degrees of educational attainment, $\chi2$=1.47, $p$=.69, did not differ between the two groups. Nor did they differ in estimated general cognitive ability (G), as measured through the Matrices subtest from WAIS-IV, $t(39)$=0.26, $p$=.79, $d$=0.08. In Fig 1, the average audiograms for both groups, based on air-conduction pure-tone audiometry, are presented.

**Table 2. Participant demographic data, hearing thresholds (PTA-4), and scaled points on the Matrices subtest from WAIS-IV.**

| | | Hearing loss | Normal hearing | | |
|---|---|---|---|---|---|
| | | M (SD) | M (SD) | t | p |
| Age (years) | | 63(8) | 52 (8) | 4.42 | <.001 |
| Matrices-WAIS-IV* | | 12 (2) | 12 (2) | 0.26 | .80 |
| PTA-4 of better ear | | 41 (9) | 5 (5) | 12.05 | <.001 |
| PTA -4 of worse ear | | 45 (10) | 8 (5) | 10.89 | <.001 |
| | | n | n | Chi2 | p |
| Gender | | | | 0.51 | .48 |
| | Female | 16 | 17 | | |
| | Male | 5 | 3 | | |
| Education | | | | 2.79 | .43 |
| | Secondary school | 4 | 6 | | |
| | College undergraduate | 4 | 4 | | |
| | College graduate | 11 | 7 | | |
| | Postgraduate studies | 2 | 3 | | |
| Hearing-aid use | An hour or less per day | 1 | N/A | – | – |
| | One to two hours per day | 2 | N/A | | |
| | Several hours per day | 2 | N/A | | |
| | Most of the day | 16 | N/A | | |

*scaled points.

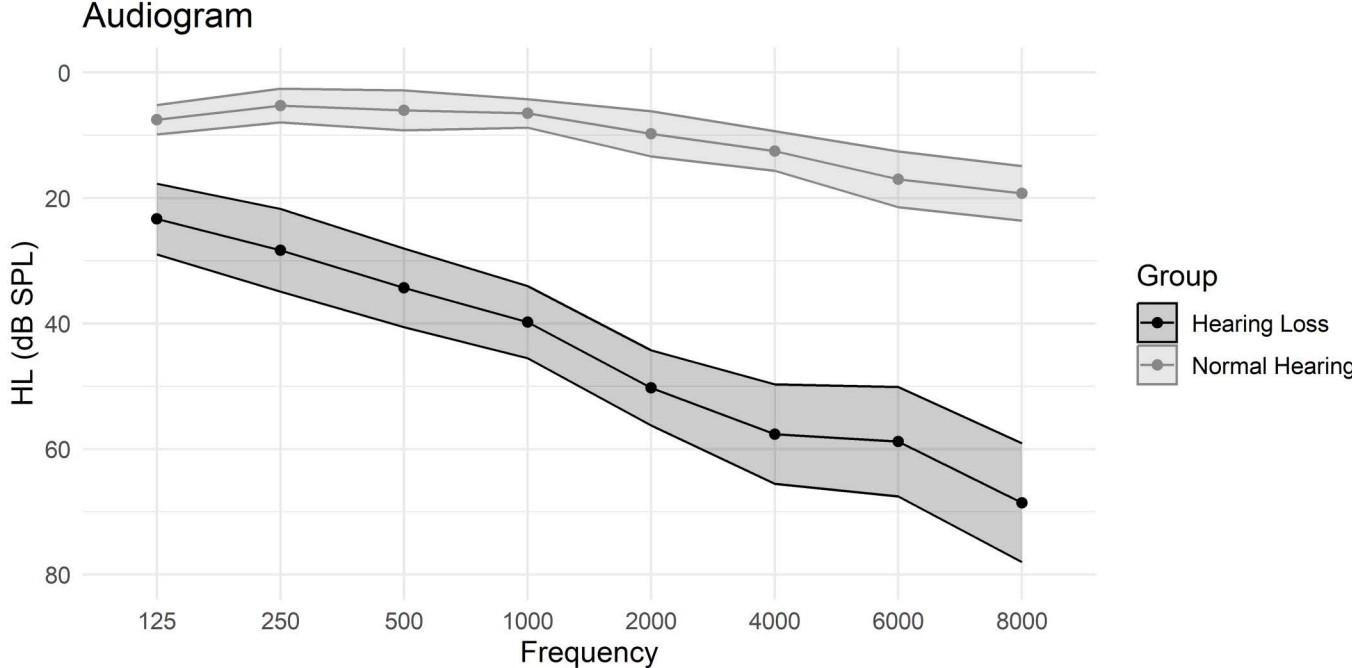

**Fig 1. Audiograms for the normal hearing group and for the hearing loss group.** Mean air conduction thresholds for the normal hearing and normal hearing loss group, including 95% confidence intervals. HL = Hearing Level.

## 3.2. Acoustic characteristics of emotions in sentences

In the sentences (Table 3), anger was characterized by a narrow formant 1 bandwidth, high loudness, and high alpha ratio (more energy in lower frequencies). Happiness was characterized by a high pitch and high formant 1–3 frequencies. Fear was characterized by low shimmer, high HNR, and high spectral slope 0–500 Hz. Sadness was characterized by low formant 1–3 relative energies (low in comparison to the other emotions). Lastly, surprise was characterized by loudness, and low HNR, low shimmer and low Spectral Slope 0–500 and 500–1500 (low relative to the other emotions).

## 3.3. Acoustic differences between emotions in sentences

Findings from PCA showed that frequency-related and amplitude-related parameters generated one component each, while spectral-balance-related parameters generated two components. The frequency component comprised high values for pitch, Frequency Formants 1, 2, and 3, and low values for the Bandwidth of Formant 1. The amplitude component comprised high values for shimmer and loudness, and low values for HNR. From the spectral-balance-related parameters, the first component comprised high values for the amplitude of Formants 1, 2, and 3, while the second component comprised high values for Alpha Ratio, Spectral slope 0–500 Hz, Spectral slope 500–1500 Hz, and low values for Hammarberg index. Values for the four components, divided by emotion, are presented in Table 4.

To determine which emotions are the most distinct based on acoustic parameters, we calculated the differences between each emotion and all other emotions (first part of prediction 3, Table 5). The analysis revealed that surprise is the most distinct emotion. Additionally, we conducted a similar analysis focusing on frequency parameters, which showed that happiness is primarily distinguished by these frequency parameters (second part of prediction 3).

The multinominal regression analyses indicated that the components significantly contributed to the prediction of emotions ($\chi^2 = 64.37$, $p < 0.001$, Nagelkerke $R^2 = 0.34$). When the frequency

**Table 3. Means of Z-scores (using each speaker's neutral sentences' means and standard deviations as reference) for acoustic parameters for different emotions expressed verbally, measured across speakers, divided by acoustic features (acoustic domains). Standard deviations presented within brackets.**

| Acoustic feature | Acoustic parameter | Anger | Happiness | Fear | Sadness | Surprise |
|---|---|---|---|---|---|---|
| *Frequency* | Pitch | 5.00 (5.39) | 7.18 (6.25) | 5.81 (2.31) | 3.99 (5.36) | 3.56 (4.14) |
| | Jitter | -0.13 (1.42) | 0.58 (2.02) | -0.98 (1.75) | 0.32 (2.63) | 2.14 (2.86) |
| | Frequency – formant 1 | 0.78 (2.58) | 1.75 (2.30) | 1.47 (1.45) | 0.12 (1.95) | 0.57 (1.02) |
| | Frequency – formant 2 | 1.20 (2.67) | 1.94 (2.10) | 1.75 (1.52) | 0.23 (2.23) | 1.03 (0.96) |
| | Frequency – formant 3 | 0.80 (2.52) | 1.58 (2.32) | 0.88 (1.20) | -0.10 (2.17) | 0.72 (0.99) |
| | Bandwidth – formant 1 | -1.05 (1.29) | -0.96 (0.95) | -0.44 (0.94) | -0.89 (1.35) | -0.82 (0.88) |
| *Energy/Amplitude/Intensity* | Shimmer | -1.03 (1.03) | -1.02 (1.47) | -1.43 (0.85) | -1.02 (0.95) | 0.13 (1.58) |
| | Loudness | 7.16 (4.32) | 6.49 (3.33) | 5.08 (5.36) | 2.96 (3.18) | 1.24 (2.16) |
| | Harmonics-to-noise ratio | 2.36 (2.45) | 3.99 (3.65) | 4.83 (1.96) | 2.16 (3.52) | 1.31 (2.91) |
| *Spectral (balance)* | Alpha ratio | 2.52 (2.73) | 2.15 (2.62) | 1.14 (1.63) | 1.95 (2.60) | 0.48 (1.56) |
| | Hammarberg index | -1.57 (1.91) | -1.20 (1.46) | -0.74 (1.60) | -1.40 (1.89) | -0.35 (1.18) |
| | Spectral slope 0–500 Hz | 2.53 (1.96) | 2.68 (2.27) | 4.90 (3.52) | 2.76 (2.61) | 1.80 (1.73) |
| | Spectral slope 500–1500 Hz | 1.45 (2.02) | 1.57 (1.90) | 1.28 (1.72) | 0.35 (1.94) | 0.12 (1.47) |
| | Relative energy – formant 1 | -0.30 (1.15) | -0.31 (1.42) | -0.19 (1.21) | -0.49 (1.36) | -0.85 (1.16) |
| | Relative energy – formant 2 | 0.32 (1.08) | 0.43 (1.03) | 0.21 (1.20) | 0.10 (1.27) | -0.56 (1.10) |
| | Relative energy – formant 3 | 0.34 (1.10) | 0.46 (1.01) | 0.24 (1.15) | 0.14 (1.30) | -0.52 (1.15) |

Note. Number of recordings in total =108; N with expression of anger=24, happiness=25, fear=22, sadness=21, surprise=16.

**Table 4. Means (standard deviations) for the four components describing acoustics of sentences, divided by emotion.**

|  | Frequency | Amplitude | Spectral balance 1 | Spectral balance 2 |
|---|---|---|---|---|
| Anger | 0.00 (1.30) | 0.16 (0.75) | 0.31 (1.16) | 0.16 (0.91) |
| Happiness | 0.40 (1.12) | 0.34 (1.06) | 0.19 (1.03) | 0.22 (0.95) |
| Fear | 0.11 (0.55) | 0.50 (0.75) | 0.02 (0.53) | 0.12 (1.00) |
| Sadness | -0.38 (1.08) | -0.17 (0.88) | 0.03 (1.14) | -0.02 (1.08) |
| Surprise | -0.15 (0.50) | -0.83 (0.97) | -0.58 (0.74) | -0.50 (0.95) |

**Table 5. Differences between emotions expressed as differences of components' mean values. The TOTAL section describes the total difference between emotions when considering all four components (by Euclidean distances).**

|  | Anger | Happiness | Fear | Sadness | Surprise |
|---|---|---|---|---|---|
| **TOTAL** |  |  |  |  |  |
| Anger | – |  |  |  |  |
| Happiness | 0.45 | – |  |  |  |
| Fear | 0.46 | 0.38 | – |  |  |
| Sadness | 0.61 | 0.97 | 0.85 | – |  |
| Surprise | 1.50 | 1.66 | 1.61 | 1.04 | – |
| **Frequency** |  |  |  |  |  |
| Anger | – |  |  |  |  |
| Happiness | 0.39 | – |  |  |  |
| Fear | 0.11 | 0.28 | – |  |  |
| Sadness | 0.38 | 0.78 | 0.49 | – |  |
| Surprise | 0.15 | 0.55 | 0.26 | 0.23 | – |
| **Amplitude** |  |  |  |  |  |
| Anger | – |  |  |  |  |
| Happiness | 0.17 | – |  |  |  |
| Fear | 0.34 | 0.16 | – |  |  |
| Sadness | 0.34 | 0.51 | 0.68 | – |  |
| Surprise | 0.99 | 1.17 | 1.33 | 0.65 | – |
| **Spectral balance 1** |  |  |  |  |  |
| Anger | – |  |  |  |  |
| Happiness | 0.13 | – |  |  |  |
| Fear | 0.29 | 0.17 | – |  |  |
| Sadness | 0.28 | 0.15 | 0.01 | – |  |
| Surprise | 0.89 | 0.76 | 0.60 | 0.61 | – |
| **Spectral balance 2** |  |  |  |  |  |
| Anger | – |  |  |  |  |
| Happiness | 0.06 | – |  |  |  |
| Fear | 0.04 | 0.10 | – |  |  |
| Sadness | 0.18 | 0.24 | 0.14 | – |  |
| Surprise | 0.66 | 0.72 | 0.62 | 0.48 | – |

Values for the differences between each emotion and all other emotions in Total; Anger = 3.02, Happiness = 3.46, Fear = 3.30, Sadness = 3.47, Surprise = 5.81, and for the Frequency component; Anger = 1.03, Happiness = 2.00, Fear = 1.14, Sadness = 1.88, Surprise = 1.19.

component alone contributed to discrimination, it was specifically for distinguishing happiness from both anger and sadness. In both cases, increased values on the frequency component increased the odds for happiness (B = 0.659, $p$ = 0.050, OR = 1.93 and B = 1.081, $p$ = 0.004, OR = 2.95, respectively). In the same way, the amplitude component alone contributed to distinguishing fear from both happiness and surprise. Increased values on the amplitude component increased the odds for fear (B = 0.825, $p$ = 0.045, OR = 2.28 and B = 1.846, $p$ < 0.001, OR = 6.34, respectively). The contribution of spectral-balance components was only present when used in combination with frequency or amplitude components. In general, higher values for spectral-balance components contributed to discrimination of anger and sadness from both fear and surprise (Spectral-Balance 1) and happiness from surprise (Spectral-Balance 2). Notably, surprise could always be distinguished (Table 6, see). For further details see SI Appendix in Supporting information.

### 3.4. Acoustic characteristics of non-verbal vocalizations

In Table 7, the means and standard deviations of z-scores for all acoustic parameters for the different non-verbal vocalizations is presented. With reference to which parameters have the highest or lowest z-scores for each emotion, in relation to the average for all other emotions, anger is characterized by low pitch, low formant 1–3 frequencies, low shimmer, and high loudness. Happiness is characterized by high jitter, high shimmer, a low Harmonics-to-Noise ratio, low Spectral slope 0–500 Hz, and high Spectral slope 500–1500 Hz. Fear is characterized by low jitter, a narrow formant 1 bandwidth, high Spectral slope 0–500 Hz, and low Spectral slope 500–1500 Hz. Sadness is characterized by high pitch, high formant 1 frequency, high formant 2 frequency (as is interest however which has the same value), and high alpha ratio.

**Table 6. Odds ratios for significant coefficients in multinomial logistic regression models with each emotion as the reference emotion, using the four components from the PCA analyses as independent variables (only significant odds ratios are shown).**

| Reference | Emotion | Frequency | Amplitude | Spectral-Balance 1 | Spectral-Balance 2 |
|---|---|---|---|---|---|
| Anger | Happiness | 1.93 | | | |
| | Fear | | 2.55 | 0.43 | |
| | Sadness | | | | |
| | Surprise | | 0.40 | 0.36 | |
| Happiness | Anger | 0.52 | | | |
| | Fear | | 2.28 | | |
| | Sadness | 0.34 | | | |
| | Surprise | | 0.36 | | 0.50 |
| Fear | Anger | | 0.39 | 2.35 | |
| | Happiness | | 0.44 | | |
| | Sadness | | 0.31 | 2.25 | |
| | Surprise | | 0.16 | | |
| Sadness | Anger | | | | |
| | Happiness | 2.95 | | | |
| | Fear | | 3.24 | 0.45 | |
| | Surprise | 3.08 | | 0.37 | |
| Surprise | Anger | | 2.49 | 2.79 | |
| | Happiness | | 2.78 | | 2.01 |
| | Fear | | 6.34 | | |
| | Sadness | 0.32 | | 2.67 | |

Note: For example, a one unit increase in the frequency component is associated with a 93% increase in the odds that an utterance was Happy rather than Angry (OR = 1.93), while at the same time also associated with a 48% decrease in the odds that an utterance was Anger rather than Happy (OR = 0.52).

**Table 7. Means of Z-scores (using all of each speaker's non-verbal vocalizations' means and standard deviations as reference) for acoustic parameters for different non-verbal vocalizations, measured across speakers, divided by acoustic features (acoustic domains). Standard deviations are presented within brackets.**

| Acoustic feature | Acoustic parameter | Anger | Happiness | Fear | Sadness | Interest |
|---|---|---|---|---|---|---|
| *Frequency-related* | Pitch | -0.84 (0.91) | -0.24 (1.16) | 0.34 (0.91) | 1.28 (1.03) | -0.39 (0.65) |
| | Frequency – formant 2 | -0.78 (1.11) | 0.13 (0.60) | -0.25 (2.10) | 0.39 (0.95) | 0.39 (0.28) |
| | Frequency – formant 3 | -0.51 (0.68) | -0.28 (0.35) | -0.34 (1.69) | 0.08 (0.55) | 2.24 (1.60) |
| | Bandwidth – formant 1 | 0.72 (0.75) | -0.36 (1.41) | -0.51 (0.26) | -0.21 (1.27) | 0.95 (1.09) |
| *Energy/Amplitude/Intensity-related* | Shimmer | -0.65 (0.44) | 2.82 (2.02) | -0.51 (0.35) | -0.30 (0.21) | -0.30 (1.31) |
| | Loudness | 2.06 (3.57) | -0.35 (0.52) | 0.12 (0.53) | -0.26 (0.58) | -0.71 (0.19) |
| | Harmonics-to-noise ratio | -0.62 (0.96) | -0.75 (0.96) | 0.32 (1.13) | 0.46 (0.68) | 0.74 (1.14) |
| *Spectral (balance)-related* | Alpha ratio | 0.24 (0.41) | 0.18 (1.53) | 0.43 (0.56) | 0.46 (0.81) | -1.88 (0.83) |
| | Hammarberg index | -0.34 (0.85) | -0.14 (0.83) | -0.60 (0.56) | -0.52 (0.69) | 2.84 (0.66) |
| | Spectral slope 0–500 Hz | -0.11 (0.62) | -1.12 (1.29) | 0.91 (1.06) | 0.74 (1.06) | -0.48 (0.38) |
| | Spectral slope 500–1500 Hz | -0.10 (0.96) | 0.53 (1.22) | -0.24 (0.80) | 0.15 (1.32) | -0.09 (1.07) |
| | Relative energy – formant 1 | 0.70 (0.77) | -0.27 (1.04) | -1.00 (0.58) | -0.31 (1.56) | 1.01 (0.04) |
| | Relative energy – formant 2 | 0.63 (0.91) | -0.44 (1.16) | -0.40 (0.54) | -0.38 (1.92) | 0.74 (0.11) |
| | Relative energy – formant 3 | 0.57 (0.90) | -0.38 (1.23) | -0.40 (0.55) | -0.43 (1.86) | 0.68 (0.10) |

Note: Number of recordings in total =20; N with expression of anger=4, happiness=4, fear=4, sadness=4, interest=4.

Finally, interest is characterized by a high formant 3 frequency, a broad formant 1 bandwidth, low loudness, high Harmonics-to-Noise ratio, low alpha ratio, high Hammarberg index, and high formant 1–3 relative energies.

### 3.5. Recognition accuracy

Descriptive data for recognition accuracy for different emotions, for both stimulus types, and with the hearing loss group listening with as well as without amplification is presented in Table 8. Included in the table is also Rosenthal's pi (47) for the sake of comparison with other studies which included different numbers of response options. No participant aborted any trials in the emotion recognition task, therefore, there is no missing data.

### 3.6. Effects of hearing loss

All main effects and interaction effects from the statistical behavioural analyses are summarized in S1 Table in Supporting information. For the analyses of sentences when the hearing loss group listened without amplification, there was a significant main effect of Emotion, $F(4,152)=2.69$, $p=.033$, $\eta^2_p=.07$. Pairwise comparisons showed that accuracy for sadness and surprise was higher than for anger, fear, happiness, all $p <0.001$. There was a significant main effect of Group, $F(1,38)=15.24$, $p <0.001$, $\eta^2_p=.29$. Accuracy measured across emotions was 18 percentage points higher for the normal hearing group. There was a significant interaction between Emotion and Group, $F(4, 152)= 3.68$, $p=0.007$, $\eta^2_p=.09$. Pairwise comparisons showed that the hearing loss group had significantly lower recognition accuracy compared to the normal hearing group for fear, $(p=.001)$, surprise, $(p=0.011)$, and happiness, $(p <0.001)$, but not for anger $(p=0.469)$, or sadness $(p=0.245)$. For the hearing loss group recognition of fear was lower compared to recognition of sadness $(p <0.001)$, recognition of happiness was significantly lower than recognition of anger $(p=0.022)$, sadness $(p <0.001)$, and surprise $(p <0.001)$, while for the normal hearing group recognition of happiness was significantly lower than recognition of surprise $(p=.004)$. There was no significant effect of the covariate of Age, $F(1,38)=0.31$, $p=0.578$. $\eta^2_p=.009$, but there was a

**Table 8. Proportions of accuracy for different emotions, divided by group, listening condition and type of emotion expression. Means (standard deviations) and Rosenthal's proportion index (Π) are presented in separate columns.**

| | Hearing loss | | | | Normal hearing | |
| | Non-amplified | | Amplified | | Non-amplified | |
| | M (SD) | Π | M (SD) | Π | M (SD) | Π |
|---|---|---|---|---|---|---|
| *Verbal* | | | | | | |
| Anger | .50 (.20) | .80 | .54 (.19) | .82 | .61 (.20) | .86 |
| Happiness | .34 (.16) | .67 | .52 (.20) | .81 | .63 (.14) | .87 |
| Fear | .46 (.23) | .77 | .53 (.14) | .82 | .67 (.19) | .89 |
| Sadness | .68 (.15) | .89 | .60 (.21) | .86 | .78 (.16) | .93 |
| Surprise | .68 (.24) | .89 | .76 (.17) | .93 | .82 (.17) | .95 |
| Overall | .54 (-) | – | .59 (-) | – | .70 (-) | – |
| *Non-verbal* | | | | | | |
| Anger | .55 (.20) | .83 | .65 (.20) | .88 | .89 (.13) | .97 |
| Happiness | .93 (.14) | .98 | .95 (.10) | .99 | .96 (.09) | .99 |
| Fear | .61 (.37) | .86 | .80 (.23) | .94 | .89 (.15) | .97 |
| Sadness | .67 (.27) | .89 | .70 (.19) | .90 | .93 (.14) | .98 |
| Interest | .80 (.25) | .94 | .94 (.13) | .98 | .97 (.08) | .99 |
| Overall | .71 (-) | – | .81 (-) | – | .93 (-) | – |

Π= Rosenthal's proportion index.

significant interaction between Emotion and Age, $F(4,152)=2.45$, $p=0.048$, $\eta^2_p=.06$. Because of this interaction we also performed the analysis without age as a covariate. In this analysis, there were still main effects of Group, $F(1, 39)=19.54$, $p<0.001$, $\eta^2_p=.33$, and Emotion, $F(4, 156)=22.38$, $p<0.001$, $\eta^2_p=.37$, but the interaction between Emotion and Group, $F(4, 156)=2.32$, $p=0.059$, $\eta^2_p=.06$, was no longer significant. This was most likely due to the group difference for sadness being larger, $p=0.048$, a 10-percentage point difference when not controlling for age compared to 7 percentage points when controlling for age. Similarly, the difference for anger was larger, $p=0.10$, with an 11 percentage point difference when not controlling for age compared to 7 percentage points when controlling for age. In summary, the HL group had poorer recognition of emotions expressed in sentences compared to the NH group, regardless of whether they listened with or without the aid of linear amplification.

For sentences, when comparing the normal hearing group with the hearing loss group using linear amplification, there was a significant main effect of Group, $F(1, 38)= 5.62$, $p=0.023$, $\eta^2_p=.13$. Accuracy was 10 percentage points higher for the normal hearing group. There was also a significant main effect of Emotion, $F(4, 152)=2.49$, $p=0.045$, $\eta^2_p=.06$. Accuracy measured across groups for surprise was significantly higher than for anger, fear, happiness (all $p<.001$), and sadness ($p=0.028$) accuracy for sadness was higher than for anger ($p=0.006$) and happiness ($p=0.025$). There was no significant interaction between Group and Emotion, $F(4,152)=0.90$, $p=0.464$, $\eta^2_p=.023$, and no significant effect of the covariate Age, $F(1,38)=0.30$. $p=0.585$, $\eta^2_p=.008$. The differences between the normal hearing and the hearing loss groups for different emotions expressed in sentences, when not controlling for age, are illustrated in Fig 2A and Fig 2B. In sum, similarly to when the hearing loss group listened without linear amplification, they exhibited poorer overall recognition of emotions expressed in sentences when using linear amplification. However, the overall difference in recognition between the two groups was smaller, and there were no significant differences in accuracy for specific emotions.

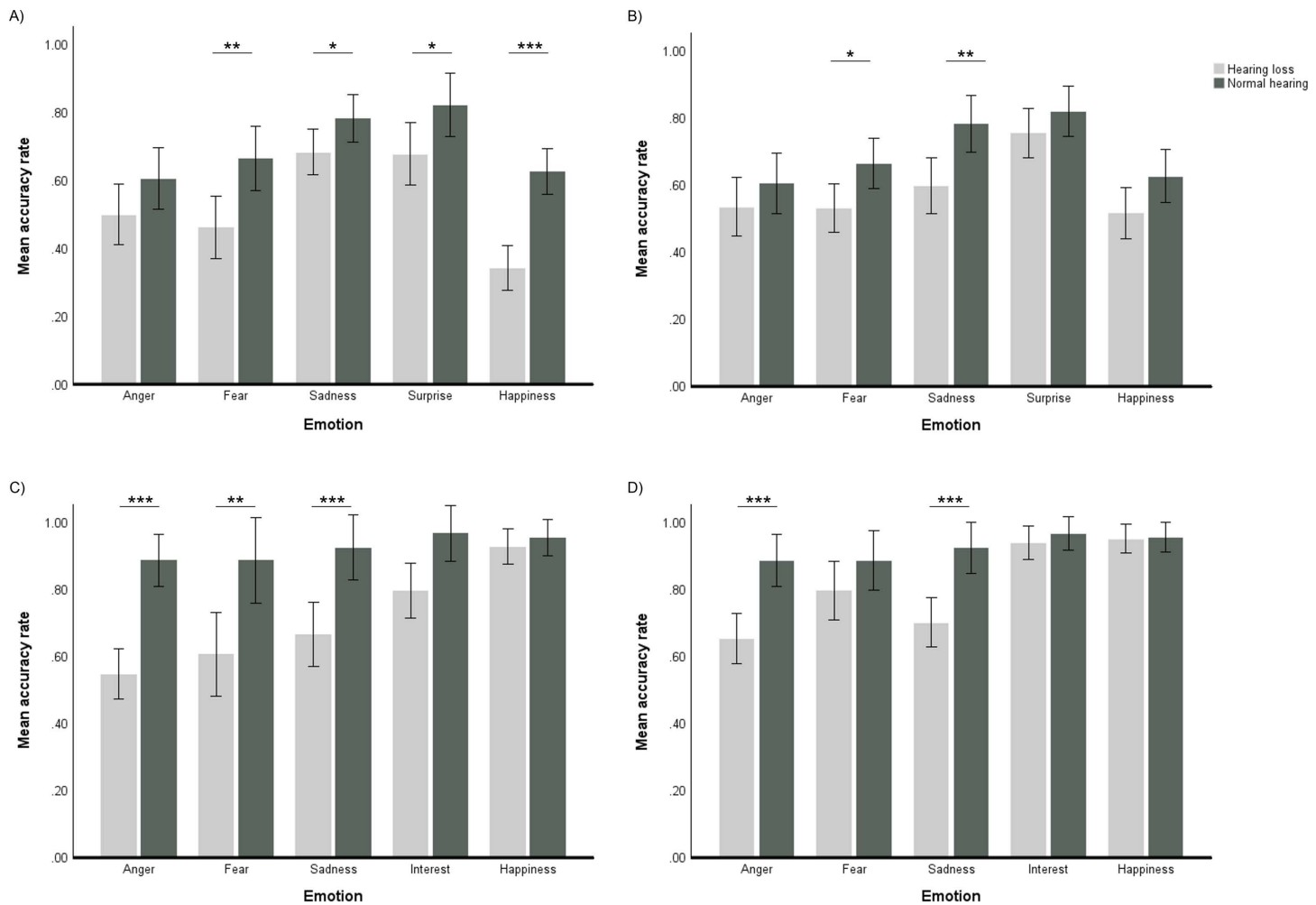

**Fig 2. Recognition of emotions expressed in sentences and non-verbal vocalizations for the hearing loss group and for the normal hearing group.** Recognition of different emotions for both groups, with the hearing loss group (HL) listening to: A) emotions expressed in sentences without amplification; B) HL listening to emotions expressed in sentences with linear amplification; C) HL listening to non-verbal vocalizations without linear amplification, and; D) HL listening to non-verbal vocalizations with linear amplification Significant differences between groups (pairwise comparisons): *p <0.05, ** p <0.01, ***p <0.001. Error bars represent 95% Confidence Intervals.

For the analyses of non-verbal vocalizations when the hearing loss group listened without amplification, there was no significant main effect of Emotion, F(3.84, 146.06)=2.05, $p$=0.090, $\eta^2_p$=.05, but there was a significant main effect of Group, F(1,38)= 25.18, $p$<0.001, $\eta^2_p$=.40. Accuracy was 22 percentage points higher for the normal hearing group. There was a significant interaction between Emotion and Group, F(3.84, 146.06)= 2.59, $p$=0.041, $\eta^2_p$=.06. The hearing loss group had significantly lower recognition accuracy for anger ($p$<0.001), fear ($p$=0.003), sadness ($p$=0.022), and interest ($p$<0.001), but not for happiness ($p$=0.264). In the hearing loss group, recognition of happiness, was significantly higher compared with accuracies for anger ($p$<0.001), fear ($p$<001), sadness ($p$=0.005), and interest ($p$=0.025), while recognition accuracy of interest was significantly higher than anger ($p$=0.016) and fear ($p$=0.036). In contrast, there were no significant differences in recognition accuracy between emotions for the normal hearing group. The covariate of Age was not significant, F(1,38)=0.15, $p$=0.704, $\eta^2_p$=.004. The interaction between Emotion and Age was nearly significant, F(3.84, 146.06)=2.451, $p$=0.051, $\eta^2_p$=.06. To explore the meaning of this nearly significant interaction,

we again performed the analysis without age as a covariate. The differences between groups for different emotions differed depending on whether age was included as a covariate or not. For example, when controlling for age recognition accuracy for interest was 24 percentage points lower for the HL group, but 17 percentage points lower when not controlling for age, and accuracy for sadness was 19% lower for the HL group when controlling for age, but 26 percentage points lower when not controlling for age. In summary, the HL group had poorer recognition of all specific nonverbal emotion expressions, except happiness, when listening without linear amplification compared to the NH group.

For non-verbal vocalizations when the hearing loss group listened with linear amplification, the main effect of Group was significant, $F(1, 38)= 10.30$, $p=0.003$, $\eta^2_p=.21$, accuracy was 10 percentage points higher for the normal hearing group compared to the hearing loss group. The main effect of Emotion was not significant, $F(3.74, 141.97)=0.76$, $p=0.547$, $\eta^2_p=.020$. The interaction between Emotion and Group was significant, $F(3.74, 141.97)= 2.80$, $p=0.031$, $\eta^2_p=.07$. Pairwise comparisons showed that the hearing loss group had significantly lower recognition accuracy compared to the normal hearing group for anger ($p=0.014$) and sadness ($p=0.002$), but not for fear ($p=0.180$), interest ($p=0.527$) or happiness ($p=0.857$). Similar to what was the case for non-amplified listening, recognition accuracy of happiness for the hearing loss group was significantly higher than accuracy of anger ($p<0.001$) fear ($p=0.026$) and sadness ($p<0.001$), while recognition accuracy for interest was significantly higher than for anger ($p<0.001$), fear ($p=0.025$), and sadness ($p<0.001$). There were no significant differences in recognition accuracy between emotions for the normal hearing group. Neither the effect of the covariate of Age, $F(1,38)=0.65$, $p=0.425$, $\eta^2_p=.02$, nor the interaction between Emotion and Age, $F(3.74,141.97)= 1.06$, $p=0.376$, $\eta^2_p=.03$, were significant. The differences between the groups for different non-verbal vocalizations, without age as a covariate are presented in Fig 2C and Fig 2D. Thus, the HL group had poorer overall recognition of non-verbal vocalizations when listening with linear amplification, as was the case for listening without linear amplification. However, in contrast to when listening without linear amplification, the HL group did not have significantly poorer recognition of fear or interest when listening with linear amplification.

## 3.7. Effects of amplification

For the analyses of the effects of linear amplification on recognition of emotions in sentences, the main effect of Listening condition was not significant, $F(1, 20)=3.10$, $p=0.093$, $\eta^2_p=.13$. The main effect of Emotion was significant, $F(4, 80)=18.27$, $p <0.001$, $\eta^2_p=.48$. Pairwise comparisons showed that recognition accuracy for surprise measured across listening conditions was higher compared to, anger, fear, and happiness (all $p <0.001$), while accuracy for sadness was higher than for anger ($p=0.014$), fear ($p=0.008$), and happiness ($p <0.001$). The interaction between Emotion and Listening condition was significant, $F(4, 80)=4.31$, $p=0.003$, $\eta^2_p=.18$. Pairwise comparisons showed that linear amplification significantly improved recognition accuracy for happiness ($p<0.001$, with a nearly 18-percentage point improvement), but not for anger ($p=0.535$), fear ($p=0.204$), sadness ($p=0.102$), or surprise ($p=0.153$). The effect of amplification on emotions expressed in sentences is presented in Fig 3A.

For the analyses of the effects of linear amplification on the recognition of nonverbal vocalizations, the main effect of Listening condition was significant, $F(1, 20)=16.80$, $p <0.001$, $\eta^2_p=.46$, there was a 10-percentage point improvement in recognition accuracy with linear amplification. The main effect of Emotion was also significant, $F(4, 80)= 14.20$, $p<0.001$, $\eta^2_p=.42$. Pairwise comparisons showed that accuracy for happiness was significantly higher than accuracy for fear ($p=0.014$), anger and sadness (both $p <0.001$), and that accuracy for interest was significantly higher than for anger ($p <0.001$), fear ($p=0.009$), and sadness

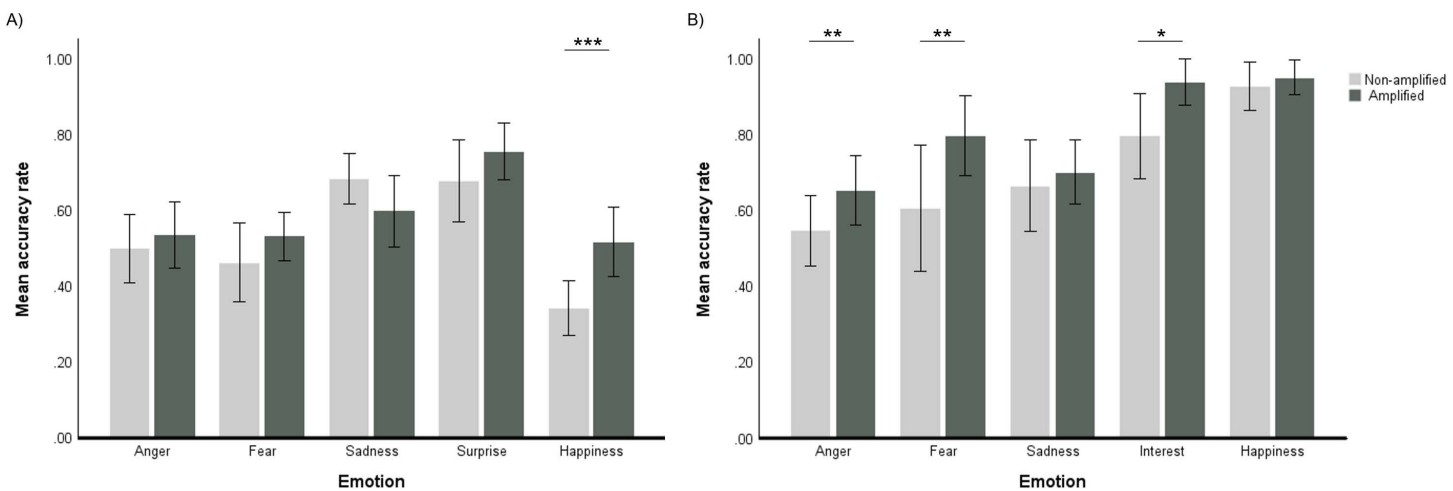

**Fig 3. Effects of amplification on emotion recognition for the hearing loss group.** Recognition of emotions by the hearing loss group listening with and without amplification for: A) Emotion expressed in sentences, and; B)Non-verbal vocalizations. Significant differences between listening conditions (pairwise comparisons): *p <0.05, ** p <0.01, ***p <0.001. Error bars represent 95% Confidence Intervals.

($p$=0.011). The interaction between Emotion and Listening condition was significant, $F(4, 80)$=2.66, $p$=0.038, $\eta^2_p$=.12. Pairwise comparisons revealed that linear amplification significantly improved recognition accuracy for anger ($p$=0.004), fear ($p$=0.004), and interest ($p$=0.010), but not for sadness ($p$=0.480) or happiness ($p$=0.428). The effect of linear amplification on the recognition of emotions expressed non-verbally are presented in Fig 3B. In summary, for the HL group, linear amplification significantly improved the recognition of happiness expressed in sentences, the non-verbal vocalizations of anger, fear, and interest.

### 3.8. Differences in confusion matrices

Examining the confusion matrices, presented in Fig 4, we note that the patterns of confusion were similar between the two groups although the hearing loss group more frequently misrecognized different emotions. The most common confusion, for both groups, pertaining to emotions expressed verbally, was the confusion of happiness for surprise, 34% without amplification and 31% with amplification, respectively for the hearing loss group, and 23% for the normal hearing group. For non-verbal vocalizations the most common confusion, for both groups, was the confusion of sadness for fear, 26% without and 27% with amplification, respectively for the hearing loss group, and 7% for the normal hearing group. Thus, the difference between the groups in the study sample regarding confusions was more quantitative than qualitative, and the tendency was for linear amplification to reduce the frequency of confusions but not the patterns.

## 4. Discussion

In this study, we analyzed the effects of mild-to-moderate hearing loss and linear amplification on the recognition of emotions expressed in sentences and non-verbal expressions in middle-aged to older individuals. Overall, we found support for some of our predictions (for an overview, see Table 9). Individuals with mild-to-moderate hearing loss had poorer general emotion recognition, regardless of the use of linear amplification, for both sentences and non-verbal emotion expressions compared to individuals with normal hearing. However, the use of linear amplification had a facilitative effect on emotion recognition, especially for

**Hearing loss, without amplification**   **Hearing loss, with amplification**   **Normal hearing**

Sentences

**a) Hearing loss: Sentences, without amplification**

| Expressed emotion | Identified emotion | | | | |
|---|---|---|---|---|---|
| | Anger | Happiness | Fear | Sadness | Surprise |
| Anger | **50** | 11 | 5 | 7 | 27 |
| Happiness | 10 | **35** | 11 | 10 | 34 |
| Fear | 10 | 4 | **46** | 20 | 20 |
| Sadness | 3 | 4 | 12 | **68** | 13 |
| Surprise | 2 | 21 | 4 | 6 | **68** |

**b) Hearing loss: Sentences, with amplification**

| Expressed emotion | Identified emotion | | | | |
|---|---|---|---|---|---|
| | Anger | Happiness | Fear | Sadness | Surprise |
| Anger | **54** | 7 | 10 | 4 | 25 |
| Happiness | 4 | **52** | 8 | 5 | 31 |
| Fear | 6 | 9 | **53** | 13 | 19 |
| Sadness | 4 | 5 | 17 | **60** | 14 |
| Surprise | 0 | 17 | 5 | 2 | **76** |

**c) Normal hearing: Sentences**

| Expressed emotion | Identified emotion | | | | |
|---|---|---|---|---|---|
| | Anger | Happiness | Fear | Sadness | Surprise |
| Anger | **61** | 6 | 9 | 2 | 22 |
| Happiness | 4 | **63** | 4 | 6 | 23 |
| Fear | 4 | 3 | **66** | 13 | 14 |
| Sadness | 4 | 3 | 10 | **78** | 5 |
| Surprise | 0 | 10 | 4 | 3 | **83** |

Non-verbal

**d) Hearing loss: Non-verbal, without amplification**

| Expressed emotion | Identified emotion | | | | |
|---|---|---|---|---|---|
| | Anger | Happiness | Fear | Sadness | Interest |
| Anger | **55** | 1 | 18 | 19 | 7 |
| Happiness | 1 | **93** | 1 | 4 | 1 |
| Fear | 8 | 7 | **61** | 14 | 10 |
| Sadness | 1 | 1 | 26 | **67** | 5 |
| Interest | 2 | 8 | 6 | 4 | **80** |

**e) Hearing loss: Non-verbal, with amplification**

| Expressed emotion | Identified emotion | | | | |
|---|---|---|---|---|---|
| | Anger | Happiness | Fear | Sadness | Interest |
| Anger | **65** | 0 | 17 | 17 | 1 |
| Happiness | 0 | **95** | 1 | 4 | 0 |
| Fear | 12 | 2 | **80** | 2 | 4 |
| Sadness | 2 | 1 | 27 | **70** | 0 |
| Interest | 0 | 0 | 2 | 4 | **94** |

**f) Normal hearing: Non-verbal**

| Expressed emotion | Identified emotion | | | | |
|---|---|---|---|---|---|
| | Anger | Happiness | Fear | Sadness | Interest |
| Anger | **89** | 0 | 6 | 5 | 0 |
| Happiness | 0 | **95** | 0 | 5 | 0 |
| Fear | 2 | 1 | **89** | 1 | 6 |
| Sadness | 0 | 0 | 7 | **93** | 0 |
| Interest | 0 | 3 | 1 | 0 | **96** |

**Fig 4. Confusion Matrices.** Distributions of responses for different emotions in percentages.

**Table 9. Summary of findings in relation to the predictions in the present study. The ✓ mark signifies complete support from the results for a prediction. The X signifies a complete or nearly complete lack of support for a prediction.**

| Prediction | |
|---|---|
| 1. Individuals with hearing loss will have poorer recognition compared to normal hearing individuals for emotions expressed verbally, regardless of acoustic features and regardless of the use of linear amplification, and for non-verbal vocalizations when linear amplifications is not used. | ✓ |
| 2. Individuals with and without hearing loss will not differ in accuracy for nonverbal vocalizations when linear amplification is used. | ✗ |
| 3 Vocal emotions, which are more distinct in terms of acoustic parameter measures, will be recognized with higher accuracy for both groups, but emotions that are distinguished mainly based on frequency parameters will be less accurately recognized by the hearing loss group. | ✓ |
| 4. The more distinct the frequency-related acoustic parameters are for an emotion, the better that emotion will be recognized when linear amplification is used compared to not. | ✓ |
| 5. Patterns of confusion will differ between individuals with and without hearing loss for both verbally and non-verbally expressed emotions. | ✗ |

happiness in the sentences and for the non-verbal vocalizations of anger, fear and interest. Lastly, we found patterns of confusion to be similar for both groups, regardless of amplification. The differences are thus primarily quantitative, i.e., differences in how frequently emotions are confused rather than which emotions tend to be confused with one another. An interpretation of these results is that the two groups perceive different emotions similarly, but that the degree of perceptual precision is lower for individuals with hearing loss.

We predicted that individuals with mild-to-moderate hearing loss, in general, should have poorer recognition of emotions expressed verbally, regardless of the use of linear amplification (first part of prediction 1). This prediction was supported as the hearing loss group had poorer general emotion recognition accuracy both when listening without and with amplification. The hearing loss group having poorer general recognition of emotions expressed verbally corresponds with previous research [31,33]. Singh et al. [32] showed that recognition of emotions expressed verbally, as assessed in lab-tasks, is correlated with perceived disability, which in turn is associated with emotional communication. These results are, therefore, likely relevant to everyday communication for individuals with mild-to-moderate hearing loss. However, we show here that the overall difference between the groups was smaller when the hearing loss group listened with linear amplification, showing a benefit from linear amplification. The results showed that linear amplification especially benefited the recognition of happiness. To the best of our knowledge, no previous studies have shown an improvement in recognition of emotions in speech through amplification.

For non-verbal vocalizations, we predicted that individuals with mild-to-moderate hearing loss would have poorer overall recognition when listening without amplification (second part of prediction 1), but not when listening with amplification (prediction 2). Our results only support the first prediction, not the second, as the hearing loss group had poorer recognition both when listening without and with amplification. Similar results have been shown by Legris et al. [33], i.e., individuals with mild-to-moderate hearing loss have poorer recognition of non-verbal vocalizations, even when using hearing aids.

Further, we predicted that both groups would have higher accuracy for emotions which are overall more acoustically distinct (first part of prediction 3), and specifically that recognition of emotions that are mainly distinguished by frequency-related parameters would be poorer for the hearing loss group (second part of prediction 3). For sentences, surprise was the most distinct emotion and also the most well-recognized emotion for both groups, irrespective of linear amplification. Anger, happiness and fear were the least distinct emotions, especially in relation to one another, and were also the emotions with poorest recognition accuracy. Hence, we found support for the first part of prediction 3. For non-verbal vocalizations, the normal hearing group showed high performance for all emotions, as did the hearing loss group for happiness. The low number of nonverbal vocalizations included in the study prevented us from performing the same PCA and nominal regression analyses as for the sentences. However, several of the individual acoustic parameters lend support for similar conclusions as for sentences. For example, several parameters related to happiness, the best recognized emotion for the hearing loss group, have the highest scores compared to other emotions; jitter (frequency-related), shimmer (amplitude-related) and spectral slope (spectral-balance related). For anger, the hearing loss group showed poorer performance without linear amplification, which is corroborated by anger having the lowest scores in frequency-related parameters, such as pitch, and formants 1, 2 and 3.

Regarding the effects of linear amplification, we predicted that emotions which are more distinct in terms of frequency-related parameters would show the most benefit from the use of linear amplification (prediction 4). Since no previous studies have found any significant effects of amplification, this prediction was based primarily on the consideration that linear

amplification would enable the listener to perceive those frequencies, which are otherwise inaudible to them due to elevated hearing thresholds. In accordance with this claim, happiness, which was the most distinct emotion in terms of frequency, was also the only emotion for which recognition was significantly improved by linear amplification for sentences. Especially, frequency was the only component distinguishing happiness from anger and from fear, and interestingly, the improvement in accuracy when listening to sentences with happiness was based on less confusions with anger and fear. In terms of numbers, the largest potential for improvement was a reduction in the confusion with surprise, but such an improvement was almost non-detectable. This may be explained by the fact that happiness and surprise are not distinguished by the frequency component, but by amplitude and spectral balance 2.

Further, for nonverbal vocalizations, linear amplification improved the recognition of anger, fear and interest. Although no statistical analyses could be performed due to the small amount of data, it is interesting to note that the perception of some frequency-related acoustic characteristics may be improved by using linear amplification. Regarding fear, the jitter parameter is low compared to the other emotions, while for anger and interest, the pitch is particularly low compared to other emotions in this dataset. Further studies may investigate whether modulating those frequency-related parameters influences emotion recognition in non-verbal vocalizations. The relevance of pitch has also been shown in previous studies in which pitch was related to recognition of happiness in speech [9] and to the non-verbal vocalization of anger [12]. In summary, the ability of listeners to more clearly perceive frequency related parameters, especially pitch, may explain the effects of linear amplification giving support for prediction 4.

Finally, we predicted that patterns of confusion would differ between individuals with mild-to-moderate hearing loss and individuals with normal hearing (prediction 5). No previous study on emotion recognition in this group has reported the patterns of confusion. However, we reasoned that hearing loss is not only associated with an elevated hearing threshold for different frequencies, but also has other perceptual consequences, such as altered pitch perception [25], which hypothetically might lead not only to more frequent confusions, but also to more qualitative differences in emotion perception. We found that the patterns of confusion were mostly similar between the two groups, regardless of the presence of linear amplification and the stimulus type, although confusions were more frequent in the hearing loss group. We interpret this as suggesting that mild-to-moderate hearing loss and linear amplification in relation to vocal emotion recognition are mostly associated with differences in degrees of perceptual precision, rather than qualitative perceptual differences. In summary, we found no support for prediction 5, in the sense that confusions were more frequent for the hearing loss group, but the patterns of confusion were similar between the groups.

There are several implications for the results of the present study. Firstly, we found, in line with previous studies, that individuals with mild-to-moderate hearing loss have poorer vocal emotion recognition [30–33], and that linear amplification does not restore normal overall vocal emotion recognition. As vocal emotion recognition is important for social communication, we recommend that clinicians inform individuals with hearing loss and their communication partners that they may experience difficulties in emotional communication and encourage strategies to achieve better emotion recognition performance, for example by relying on other modalities, such as vision.

As linear amplification does not normalize vocal emotion recognition, we suggest that future studies should further examine other means of improving vocal emotion recognition in individuals with hearing loss. To our knowledge, the effectiveness of vocal emotion recognition training for individuals with hearing loss has not been examined. It may be of particular

interest from a clinical perspective to examine the effectiveness of an online vocal emotion recognition training task, due to the flexibility of administration.

Additionally, the most common confusion was between happiness and surprise, despite their distinct acoustic features, which suggests that cognitive processes, beyond just acoustic processing, play a significant role in recognizing vocal emotions.

Lastly, the results regarding the effects of linear amplification found in the present study may differ from those of amplification with compression [31–34]. We suggest that such potential differences should be examined more directly by comparing the effects of linear amplification to those of amplification and compression on the recognition of different vocal emotions through controlled experiments.

## 4.1. Limitations

There are limitations to the present study including some departures from the stage 1 registered report. We were not able to include the number of participants suggested by our power analyses and it resulted in reduced power to detect small effects. The primary reason was that several individuals with self-reported normal hearing did not fulfill the strict study criteria for normal hearing and were therefore excluded. However, a sensitivity analysis performed in G*Power version 3.1.9.7 [39] indicated that we would still be able to show significance for relatively small effect sizes ($\eta_p^2 \geq .032$). To find more participants with normal hearing according to the study's criteria, we extended the age span from 50–75–40–75. Consequently, there were more normal hearing participants in the lower age span and fewer in the higher age span compared to the hearing loss group. Because of this, age was included as a covariate in analyses including group as a factor. The need to include age as a covariate, most likely further reduced the ability to identify effects. Despite this, examination of obtained effects showed that we were able to identify the effects of relevance to our predictions. With larger sample size, one could potentially have identified effects of linear amplification for more emotions. Future studies should replicate and build upon the present study by including more participants to overcome the limitations related to sample size. In summary, we diverged from the original plans for a larger sample size and age-matched groups primarily to maintain our relatively strict criteria for normal hearing.

Furthermore, the number of nonverbal vocalizations included for each emotion in the recognition task was relatively few, limiting the ability to generalize about the recognition of different emotions expressed non-verbally. Due to the limited number of nonverbal vocalizations, we were not able to perform a PCA and nominal regression analyses, preventing us from drawing conclusions about which acoustic features significantly differentiate between emotions.

Another, minor deviation from the stage 1 registered report is that audiograms were not possible to retrieve from medical journals, and instead we included tone audiograms also for individuals with hearing loss in our testing procedure.

In addition, we did not include a neutral condition in the recognition task. In the validation study [44], which was used to select stimuli for this study, neutral sentences were used to familiarize the participants with the voices of each speaker.

Moreover, the definition of sensorineural hearing loss (i.e., type of hearing loss) is based on self-reports. This makes arguments regarding the links between recognition in the hearing loss group and frequency-related acoustic characteristics of different emotions less reliable, as diminished pitch perception and frequency-selectivity are characteristic of sensorineural hearing loss. Future research should evaluate sensorineural hearing loss based on clinical diagnosis rather than self-reporting. It is important to note, however, that the participants' audiograms showed good low-frequency thresholds, suggesting that conductive losses did not

significantly affect their hearing. Additionally, their audiograms showed a pattern of sloping high-frequency hearing loss, indicating a sensorineural hearing loss of presbycusis origin.

Finally, we examined the effect of linear amplification, which is different from what is used in most hearing aids. Since hearing aids usually include different forms of compression, the dynamics can be affected, and thereby also the hearing level and perceived prosody. Typical hearing aids may affect the ability to perceive emotions expressed in speech in a negative manner compared to linear amplification. Thus, inferences about the usefulness of common hearing aids are limited.

## 5. Conclusions

Middle-aged-to-older individuals with mild-to-moderate hearing loss have poorer recognition of emotions expressed in speech and nonverbal vocalizations compared to individuals with normal hearing. Both groups tend to confuse the same emotions for one another, indicating that they perceive them similarly in a qualitative sense. Contrary to conclusions made in previous research, we show that linear amplification is effective for improving vocal emotion recognition in individuals with mild-to-moderate hearing loss. It does not impact all emotions equally, however, and should not be viewed as a sufficient solution. Importantly, we show that the emotions that were significantly improved by linear amplification were characterized by salient frequency-related characteristics. The design of the present study, however, does not allow inferences regarding whether clearer perception of these characteristics through amplification was directly related to improved recognition. Since vocal emotion recognition is important for social communication, it is important that patients and their communication partners are made aware of an increased risk of emotional miscommunication related to hearing loss. Furthermore, future studies may examine whether vocal emotion recognition in individuals with hearing loss can be improved through training.

## Supporting information

**S1 Appendix. Results from nominal regression analyses, using the four components from the PCA analyses as independent variables and different emotions as references.** (XLSX)

**S1 Table. Results from analyses of emotion recognition accuracy, Mixed ANOVAs and Repeated Measures ANOVAs.** (DOCX)

## Author contributions

**Conceptualization:** Mattias Ekberg, Örjan Dahlström, Josefine Andin, Stefan Stenfelt.

**Data curation:** Mattias Ekberg, Josefine Andin, Carine Signoret.

**Formal analysis:** Mattias Ekberg, Örjan Dahlström, Josefine Andin, Carine Signoret.

**Funding acquisition:** Örjan Dahlström.

**Investigation:** Mattias Ekberg.

**Methodology:** Mattias Ekberg, Örjan Dahlström, Stefan Stenfelt.

**Project administration:** Mattias Ekberg, Örjan Dahlström, Josefine Andin, Carine Signoret.

**Resources:** Stefan Stenfelt, Carine Signoret.

**Software:** Örjan Dahlström, Stefan Stenfelt.

**Supervision:** Örjan Dahlström, Josefine Andin, Stefan Stenfelt, Carine Signoret.

**Validation:** Mattias Ekberg, Örjan Dahlström, Josefine Andin.

**Visualization:** Stefan Stenfelt, Carine Signoret.

**Writing – original draft:** Mattias Ekberg, Örjan Dahlström, Josefine Andin, Stefan Stenfelt, Carine Signoret.

**Writing – review & editing:** Mattias Ekberg, Örjan Dahlström, Josefine Andin, Stefan Stenfelt, Carine Signoret.

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
