## [Decision Letter · Decision Letter 0]

4 Dec 2024

Dear Dr. Ekberg,

Thank you for submitting your manuscript to PLOS ONE. After careful consideration, we feel that it has merit but does not fully meet PLOS ONE’s publication criteria as it currently stands. Therefore, we invite you to submit a revised version of the manuscript that addresses the points raised during the review process.

**General Comments**

**Clarity and Structure** : The overall structure of the report is clear and logical. However, some sections could benefit from further elaboration to enhance reader understanding.

**Specific Comments**

**Introduction**

Consider providing a brief overview of the existing literature on vocal emotion recognition in individuals with hearing loss. This context will strengthen the rationale for your study.

**Methods**

Please clarify the criteria for participant selection. A more detailed description of inclusion/exclusion criteria would enhance the reproducibility of your study. It would be beneficial to specify the types of signal amplification used and how they were implemented during the vocal emotion recognition tasks.

**Results**

Clarifying how missing data were handled would be useful for interpreting the robustness of your results.

**Discussion**

In discussing your findings, consider linking them back to the specific hypotheses you outlined in the introduction. This will reinforce the contributions of your work to the field. It would be valuable to address potential implications for clinical practice or future research directions based on your findings.

**Figures and Tables**

Ensure that all figures and tables are clearly labeled and referenced in the text. Consider adding a brief description of what each figure/table illustrates.

**Minor Edits**

Proofread for any grammatical errors or typos to improve the overall professionalism of the report.

We look forward to receiving your revised manuscript.

Kind regards,

Hina Hadayat Ali, Ph.D

Academic Editor

PLOS ONE

https://journals.plos.org/plosone/article?id=10.1371/journal.pone.0261354

In your revision ensure you cite all your sources (including your own works), and quote or rephrase any duplicated text outside the methods section. Further consideration is dependent on these concerns being addressed.

Reviewers' comments:

Reviewer's Responses to Questions

**Comments to the Author**

1. Does the manuscript adhere to the experimental procedures and analyses described in the Registered Report Protocol?

Reviewer #1: Yes

Reviewer #2: Yes

Reviewer #3: Yes

Reviewer #4: Yes

Reviewer #5: Partly

2. If the manuscript reports exploratory analyses or experimental procedures not outlined in the original Registered Report Protocol, are these reasonable, justified and methodologically sound?

Reviewer #1: Yes

Reviewer #2: Yes

Reviewer #3: Yes

Reviewer #4: Yes

Reviewer #5: Yes

3. Are the conclusions supported by the data and do they address the research question presented in the Registered Report Protocol?

Reviewer #1: Yes

Reviewer #2: Yes

Reviewer #3: Yes

Reviewer #4: Yes

Reviewer #5: Partly

4. Have the authors made all data underlying the findings in their manuscript fully available?

Reviewer #1: Yes

Reviewer #2: Yes

Reviewer #3: Yes

Reviewer #4: Yes

Reviewer #5: Yes

5. Is the manuscript presented in an intelligible fashion and written in standard English?

*PLOS ONE*

Reviewer #1: Yes

Reviewer #2: Yes

Reviewer #3: Yes

Reviewer #4: Yes

Reviewer #5: Yes

Reviewer #1: It is an rigourus and detailed work.

Abstract: Without comments

Introduction: Without comments

Methods:

• When are describing the participants you report sociodemographic data, these should be reported on chapter of results.

• Although the manuscript has the title of study design, it is not reported. Instead, the authors describe the procedure for assess the outcome variables

Results:

• Table 3a. In the footnote: Note. Number of recordings in total =108; N with expression of anger=24, happiness=25, fear=22, sadness=11, surprise=16. The total of the sum of these values is 98

Discussion: Without comments

Limitation of the study: You recognized the limitations and discussed these. The main limitiation from my point of view if the groups are not comparable by the age. However you control it with different analyses. The other important limitation was the power of the achieved sample size. It is a great reason for continuos the study in a further step.

Conclusion: without comments

Reviewer #2: This study provides valuable insights into the relationship between hearing loss and emotional recognition. However, several points require attention for improvement:

- The authors should address how relying on air-conduction audiometry instead of comprehensive audiometric profiles may impact the reliability of their findings, as this is a minor deviation from the protocol.

- The reported effect size (η² = 0.02, f = 0.14) suggests the study is only powered to detect very small effects. With a sample size of N=70 participants, this raises concerns about the robustness of conclusions, especially in clinical contexts where effect sizes are typically larger (Cohen, 1988). Consequently, the findings may not generalize to the broader population of individuals with hearing loss, as significant effects in psychology and audiology often necessitate larger samples for reliable conclusions (Maxwell & Delaney, 2004).

1. Cohen J. Statistical power analysis for the behavioral sciences. 2nd ed. New Jersey, NJ: Lawrence Erlbaum Associates; 1988.

2. Maxwell SE, Delaney HD, Kelley K. Designing experiments and analyzing data: A model comparison perspective. Third edition / Scott E. Maxwell, Harold D. Delaney, and Ken Kelley. | New York,: Routledge; 2017.

- Why happiness improved specifically. Does this finding correlate with existing literature on speech perception, and could it be linked to particular frequency bands? Provide mechanistic insights or referencing previous research.

- Why non-verbal improvements do not mirror those observed in verbal expressions. Could frequency-related parameters elucidate this difference? Discuss further.

- While the similar confusion patterns between groups are noteworthy, the implications of these findings (e.g., perceptual imprecision versus qualitative differences) should be discussed further, considering how altered pitch perception may influence these findings.

- The repetition of results previously covered in the "Results" section dilutes clarity. Instead, the discussion should focus on how the findings align with or challenge previous similar studies.

Reviewer #3: Abstract:

The abstract clearly defines the importance of the research subject, highlighting the deficiency in comprehension about the impact of hearing loss on emotion detection, notwithstanding progress in hearing aid technology. Nonetheless, below is an evaluation of its areas for enhancement:

1. It may be beneficial to briefly define phrases such as "linear amplification" for readers unfamiliar with audiological terminology.

2. Incorporating precise statistics or effect sizes pertaining to enhancements in emotion recognition could enrich the findings and their importance.

Introduction:

The introduction offers a thorough review of the context and importance of the study on vocal emotion identification in individuals with hearing impairment. Here are recommendations for enhancement:

1. The flow could be improved with appropriate section organization. For instance, categorizing the sections on “Vocal Emotion Recognition” and “Effects of Hearing Loss” with specific subheadings could enhance readability, my suggestion is to add these subheadings to the introduction text accordingly, such as under “Vocal Emotion Recognition” section

• Importance of Vocal Emotion Recognition

• Acoustic Characteristics of Vocal Emotions

• The impact of auditory impairment on the perception of voice emotions

Furthermore, add this subheading under “Effects of Hearing Loss” section

• Prevalence of Hearing Loss

2. Now, the introduction does not include information regarding the demographics of the participants in the earlier studies, it is important to briefly explain (in one paragraph) the demographic data (other age groups) of individuals who participated in prior studies. This will help to contextualize the findings and ensure that they are applicable to a wider population.

3.

4. There are some typo errors e.g., “different lower--level features”, “not yielded entirelyconsistent results”, (see for example 5-8 and 16-18),. hearing loss that inloves involves, Thisis likely related to, commonly usedto characterize.

Materials and Methods:

The following are a few recommendations for enhancing the Methods section of your document:

2.1 Participants

1. Revise Inclusion/Exclusion Criteria: Clearly differentiate between inclusion and exclusion criteria

2. The p-value should be written as p<0.001 rather than p<.001

3. The sociodemographic data should be presented in the results section.

4. Table 1 Description: Summarize Table 1 with a sentence that emphasizes the significant demographic differences before its presentation. Also, the table needs to be presented in the results section.

Results:

The results section is comprehensive and provides extensive information of the findings. Here are some aspects for enhancement to contemplate for the article review:

1. The structure of the result might be improved by having clearer subsections, for example, Descriptive Data, Recognition Accuracy, Effects of Hearing Loss, and "Effects of Amplification, with titles for each subsection. The reader would be better able to navigate the findings in a methodical manner if this was done.

2. In Tables 3a and 3b, the authors may merge the columns of acoustic feature and acoustic parameter

3. All figures need improvement regarding their quality.

Discussion:

The research effectively addresses a gap in the current state of knowledge concerning the influence of linear amplification on emotion recognition. The discussion is comprehensive, confronting the limitations and relating the findings to existing research in a transparent manner. However, the authors may suggest:

1. Future research should evaluate sensorineural hearing loss based on clinical diagnosis rather than self-reporting.

2. Table 5 was not mentioned or referred to it in the discussion section.

Conclusion:

The conclusion that authors have composed is coherent and highlights the principal conclusions of your study; but it would be advantageous to adopt a more official tone, implement a little rearrangement for enhanced coherence, and incorporate a more prospective outlook. Here is a polished iteration:

1. Restructured the principal points to facilitate a more seamless transition from overarching findings to particular results and implications.

2. Incorporated a request for further research, enhancing the conclusion's forward-thinking nature and highlighting potential avenues that your work may inspire.

Reviewer #4: It is a great work. The authors discussed the problem in detailed way. It is eye-opening piece of work.

Reviewer #5: Summary of study:

In the current study, listeners with normal hearing and listeners with hearing loss heard sentences and nonverbal vocal expressions produced with 5 different emotions, and identified which emotion was presented. Listeners with HL were tested with and without hearing aid processing of stimuli, to create aided and unaided listening conditions. The authors also measured the frequency-, amplitude-, and spectrum-related characteristics of the two classes of stimuli, to explain some of the differences in listener performance. The results showed that listeners with hearing loss generally performed less accurately than listeners with normal hearing, with a larger difference when unaided (18-22 points' difference for the two types of material) than when aided (10 points' difference). The improvement in aided listening was mainly due to better recognition of "happy" in sentences, and "anger", "fear" and "interest" in nonverbal expressions. There isn't much discussion about the acoustics of different emotion categories, except to note that the acoustic distinction between emotions doesn't map well onto the perceptual distinction between emotions.

Main comment:

There have been multiple studies on hearing loss, hearing aids and vocal emotion recognition, but none of them have included a detailed investigation into the acoustics of their stimuli to explain which cues listeners with hearing loss might be using and how they might contribute to poorer performance. The most promising (and novel) contribution of this study was the investigation into the acoustic characteristics of emotional speech using a comprehensive suite of measurements, and linking the acoustics to hearing loss and hearing aid processing. However, the paper didn't really follow through on this aspect. The authors highlighted acoustics in the registered protocol, as well as in the introduction and methods of the current paper. There was a detailed table of acoustic measurements for the two classes of stimuli, but almost no integration of the large amount of acoustics results with the behavioral results. This was surprising given the authors' comments about frequency resolution being affected in sensorineural hearing loss and speculation about what cues hearing aids might be amplifying. The connection between acoustics and listeners' performance was limited to a note in the discussion about unpublished data and how the association wasn't as expected. In its current form, it's not clear what new contribution this paper brings to the literature, apart from a minor (and unexplained) finding that hearing aids help listeners to recognize happiness in spoken sentences. To help form a more coherent narrative, the authors need to: (a) strengthen the acoustics aspect to carry it through the paper, and (b) reconcile why they think looking at acoustics is helpful with the studies they cited that suggest the emotion recognition deficit isn't just (or even mainly) due to peripheral/sensory losses.

Detailed comments:

Abstract:

"making this a high-priority research topic" -- the authors could include a brief note about why, to strengthen the rationale.

Introduction:

p.5: "The average threshold, which includes several of these frequencies, is the pure-tone-average (PTA)" -- the authors could specify which frequencies, instead of "several"; there's no definition later on in the methods for PTA-4.

p.6: "Several studies have found mild-to-moderate hearing loss is associated with deficits in vocal emotion recognition (19, 27-29, but also see 30-31 for different conclusions)." -- this statement is rather vague and could be misleading for readers, implying that peripheral losses lead to emotion recognition deficits; the authors could clarify instead if these studies are referring to changes in higher-order auditory processing, which would make sense in the context of the next sentence about hearing aids not helping. Do the other studies (30, 31) find no association between hearing loss (peripheral or otherwise) and emotion recognition deficits at all?

Aims:

The first sentence is redundant, given the more specific aims that follow.

There is some inconsistency between the content of the introduction and the current study's predictions. For example:

p.6: "Modern hearing aids use non-linear amplification and compression, which involves the selective amplification of more quiet sounds" -- the authors don't explain why they chose the Cambridge linear amplification formula, either in the Study Design or earlier

p.6: "Hearing aids using linear amplification ... do not restore frequency selectivity", but prediction #5 says "The more salient the frequency-related acoustic parameters of an emotion are, the better that emotion will be identified when linear amplification is used"

Methods:

Figure 1 (audiogram) - a standard audiogram usually has the y-axis reversed

Task and study design:

There could be a little more description of the nonverbal utterances, besides their duration. Perhaps a pseudo-phonetic example (e.g. "hmm") for each emotion could be provided?

p.12 "The stimulus material is based on fourteen emotionally neutral sentences from the Swedish version of the hearing in noise test (HINT; 38), and non-verbal emotion expressions." -- It would be clearer to mention there are 20 nonverbal stimuli here, instead of presenting that information on p.15

p.12: It might also be easier for the reader to keep track of the emotion categories that were used, if there was a brief note on p.12 about which categories of emotions were dropped, instead of that information only appearing in the validation section later on p.13.

p.15 "Thus, there were in total 108 sentences (x with anger, y with happiness, z with fear, v with sadness, and w with surprise) and 20 non-verbal expressions (a with anger, b with happiness, c with fear, d with sadness, and e with interest)." -- the authors need to fill in the numbers

Acoustic analyses:

The authors could specify if the acoustics are mean-centered (not just centered). Were they normally distributed, if that matters for the z-transformation?

Prosody was mentioned in the introduction as well as the registered protocol. Which of these GeMAPS measures would reflect prosody, and could they explain some of the difference in performance between the two hearing groups?

Table 2 -- instead of leaving F0 a blank, perhaps the authors could note how F0 is calculated in GeMAPS (e.g. cross-correlation, etc)

Behavioral analyses:

What software was used for the statistical analysis? The authors could also specify which factors were between- or within-subjects, just to be clear about the different types of group comparisons.

p.18 Rosenthal's Proportion Index - it wasn't immediately clear what this note referred to, as it appeared in the section on accuracy/behavioral analysis. It would be better to move this to the earlier section on acoustic analysis since it's relevant to the stimuli - if it does refer to the acoustic measurements? A brief explanation about what effect size the index refers to would be helpful, e.g. differences between emotions.

Results:

In general, the results section could be better organized. If the authors are mainly interested in the effects of hearing loss and hearing aids, the results could be organized according to these sections, instead of separating them by the class of stimuli (verbal and nonverbal). It would also make it easier to spot overall trends across classes of stimuli.

The behavioral results may be easier to follow with tables for the main effects and interactions and associated numbers, and a narrative summary of the overall pattern of results.

More appropriate sub-headers could be used; e.g., instead of "Descriptive data" (p.18), it could be "Acoustic differences between emotions" or something like that. There is also no description of the acoustic findings for nonverbal stimuli besides the fact that they differ from the sentence stimuli.

p.17 "The same analyses were then repeated but using sessions with amplification for the hearing loss group." -- could be rephrased to improve clarity; also, the unaided to aided comparison wouldn't be a between-groups analysis but rather within-groups

I suggest being consistent about capitalizing main and interaction effects.

Table 4 - typo in "Rosenthal". Also, a brief note in the caption about what Rosenthal's Index refers to here would be helpful, e.g. difference between which groups. Are there different versions of the index for within-group (one sample) and between-group (two samples), if this is relevant?

p.28 "Difference in confusion matrices" -- these percentage values appear to refer only to one direction of Expressed  Identified cells in the matrix, e.g. "happy" identified as "surprised". But wouldn't confusion of two emotions go both ways, so that the measure of confusion should incorporate both "happy"  "surprised" and "surprised"  "happy"?

Table 3A -- can the Pitch measure be 5 to 7 SD above the mean for neutral? Is Pitch a more standard measure in the emotion acoustics literature than F0?

Which of these GeMAPS measure capture prosody?

Discussion:

p.29: "we found patterns of confusion to be similar for both groups and regardless of amplification, which suggests that the two groups perceived different emotions similarly, but that the degree of perceptual precision was lower for individuals with hearing loss." -- this statement doesn't really "suggest" anything. Rather, it shows results, which need interpretation to suggest something.

It might help to orient the reader if predictions were in the same order as before; e.g. Prediction 3 is presented last now.

The authors grouped the acoustic measurements into frequency-, amplitude- and spectral-related measurements earlier, and also mentioned that sensorineural hearing loss and hearing aid processing might affect different characteristics differently. What happened to this chain of thought?

p.31: "in the present study, recognition of happiness was poorer for the hearing loss group, and in the hearing loss group it was also, in contrast to the normal hearing group, poorer than recognition of anger and sadness. In summary, results were indicative, but not substantial, for the second part of prediction 4." -- needs some rephrasing to improve clarity

Limitations:

"Most people can be expected to be aware of their type of hearing loss" - I'm not sure that most people would know whether or not they have sensorineural hearing loss... Can the authors use their answers from their screening; e.g. reported noise exposure through occupation or leisure activities, ototoxic medications, self-reported past hearing problems etc?

Conclusions:

Table 5 should probably be located earlier, in the discussion section on predictions.

**Do you want your identity to be public for this peer review?** For information about this choice, including consent withdrawal, please see our Privacy Policy

Reviewer #1: No

Reviewer #2: No

Reviewer #3: No

Reviewer #4: No

Reviewer #5: No

---

## [Author Response · Author response to Decision Letter 1]

20 Jan 2025

Academic editor

Introduction

Comment 1. “Consider providing a brief overview of the existing literature on vocal emotion recognition in individuals with hearing loss. This context will strengthen the rationale for your study”

Response:

Thank you for your suggestion. In response, we have included additional text (pages 6-7 ) to provide an overview of the existing literature on vocal emotion recognition in individuals with hearing loss. Furthermore, we have made additional changes to the introduction. These include structural changes, detailed descriptions of previous studies on the acoustic characteristics of different emotion categories, and a rationale for examining the effects of linear amplification.

Methods

Comment 2. “”Please clarify the criteria for participant selection. A more detailed description of inclusion/exclusion criteria would enhance the reproducibility of your study.”

Response:

We have clarified the criteria for participant selection by including more detailed descriptions and dividing the text into separate sections for inclusion and exclusion criteria, as suggested by one of the reviewers. (Pages 10 and 11)

Comment 3. “It would be beneficial to specify the types of signal amplification used and how they were implemented during the vocal emotion recognition tasks”

Response:

Thank you for this valuable suggestion. We agree that specifying the types of signal amplification and implementation is beneficial. Accordingly, we have added a detailed description on page 13 :

” The stimuli were delivered through AKG K182 headphones, connected to a Supelux HA3D headphone amplifier. In the aided listening conditions, the audio files were preprocessed in MATLAB v. R2022a using an algorithm that implemented the Cambridge formula for linear hearing aids (36), based on each participant’s audiogram. These filtered audio files were then presented to individuals with hearing impairment, simulating the experience of hearing through hearing aids with the specific gain function”

Results

Comment 4.“Clarifying how missing data were handled would be useful for interpreting the robustness of your results.”

Response:

We agree with the reviewers concern regarding missing data. However, we would like to clarify that there were no instances of missing data in this study. The experimental task was designed to ensure that participants could not refrain from responding unless they chose to terminate the task prematurely. Notably, no participant opted to terminate before completion. We have added a statement about this in the text (page 32):

“No participant aborted any trials in the emotion recognition task, therefore, there is no missing data. “

Discussion

Comment 5.“In discussing your findings, consider linking them back to the specific hypotheses you outlined in the introduction. This will reinforce the contributions of your work to the field.”

Response:

Thank you for this suggestion. We have revised our discussion to more clearly link our findings to the predictions outlined in the introduction. Additionally, to maintain coherence and logial flow, we have reordered the predictions in the introduction. Specifically, prediction number 3 is now presented last, as prediction number 5. This adjustment ensures that the discussion follows a more logical and thematic sequence.

Comment 6. “It would be valuable to address potential implications for clinical practice or future research directions based on your findings”

Response:

We agree that addressing potential implication for clinical practice and future research would enhance the value of our text. As our results do not directly say anything about clinical practice, we have allowed ourselves to address potential implications more generally, by putting forth the need to raise awareness among patients and their communication partners about potential difficulties. We also discuss the insufficiency of linear amplification as a solution and suggest that future studies should explore the benefits of vocal emotion recognition training. Additionally, we suggest that future studies should directly compare the effects of linear amplification with those of amplification with compression (which is typically implemented in hearing aids), in controlled experiments.

Figures and Tables

Comment 7. “Ensure that all figures and tables are clearly labeled and referenced in the text. Consider adding a brief description of what each figure/table illustrates”

Response:

We appreciate the reviewer’s observation. We have identified that table 5 (in the revised version this table is table 9) was not referenced in the text and not described. We have added a description.

Minor Edits

Comment 8. “Proofread for any grammatical errors or typos to improve the overall professionalism of the report”

Response:

We have thoroughly reviewed the manuscript for grammatical errors and typographical mistakes and have made the necessary corrections to enhance the report.

Journal requirements

Comment 9. “Please ensure that your manuscript meets PLOS ONE's style requirements, including those for file naming. The PLOS ONE style templates can be found at

https://journals.plos.org/plosone/s/file?id=ba62/PLOSOne_formatting_sample_title_authors_affiliations.pdf”

Response:

We have updated the file names to comply with the journal’s style requirements.

Comment 10. “We noticed you have some minor occurrence of overlapping text with the following previous publication(s), which needs to be addressed: https://journals.plos.org/plosone/article?id=10.1371/journal.pone.0261354. In your revision ensure you cite all your sources (including your own works), and quote or rephrase any duplicated text outside the methods section. Further consideration is dependent on these concerns being addressed”

Response:

We have searched for overlapping text with the stage 1 registered report. The introduction has been substantially revised, so there should no longer be any such overlap.

Comment 11. “When completing the data availability statement of the submission form, you indicated that you will make your data available on acceptance. We strongly recommend all authors decide on a data sharing plan before acceptance, as the process can be lengthy and hold up publication timelines. Please note that, though access restrictions are acceptable now, your entire data will need to be made freely accessible if your manuscript is accepted for publication. This policy applies to all data except where public deposition would breach compliance with the protocol approved by your research ethics board. If you are unable to adhere to our open data policy, please kindly revise your statement to explain your reasoning and we will seek the editor's input on an exemption. Please be assured that, once you have provided your new statement, the assessment of your exemption will not hold up the peer review process”

Response:

Data from the experimental task and the acoustic analyses together with scripts and the material used in the study are publicly available on OSF. This sentence has been added to the manuscript:

“The data that support the findings of this study are openly available at the Open Science framework (OSF) at https://osf.io/, Identifier: DOI 10.17605/OSF.IO/V3ANK. Project name: “Vocal emotion identification in middle-aged-to older individuals with sensorineural hearing loss and normal hearing individuals”.

We do not have permission from the review ethics board to share personal information from the online survey. Since this information was only used for checking inclusion and exclusion criteria, it is not needed to evaluate the results of the experimental task.

Comment 12. “Please review your reference list to ensure that it is complete and correct. If you have cited papers that have been retracted, please include the rationale for doing so in the manuscript text, or remove these references and replace them with relevant current references. Any changes to the reference list should be mentioned in the rebuttal letter that accompanies your revised manuscript. If you need to cite a retracted article, indicate the article’s retracted status in the References list and also include a citation and full reference for the retraction notice”

Response:

We have reviewed our reference list, and it is complete and correct.

Reviewer #1

General

Comment 13. “It is an rigourus and detailed work.”

Response:

Thank you for this positive assessment.

Methods

Comment 14. “When are describing the participants you report sociodemographic data, these should be reported on chapter of results”

Response:

We have moved the description of sociodemographic data, including the corresponding table, to the Results section.

Comment 15. “Although the manuscript has the title of study design, it is not reported. Instead, the authors describe the procedure for assess the outcome variables”

Response:

Thank you for drawing our attention to this issue. We have made the following changes to address it: The subsection previously titled “Study design” has been renamed to “Procedure”. We have also added more detailed information regarding the implementation of linear amplification.

Results

Comment 16. “Table 3a. In the footnote: Note. Number of recordings in total =108; N with expression of anger=24, happiness=25, fear=22, sadness=11, surprise=16. The total of the sum of these values is 98”

Response:

Thank you for pointing out this error. The number of sentences with the expression of sadness was in fact 21, not 11. We have changed from 11 to 21 in the footnote of table 3a (now Table 3).

Limitation of the study

Comment 17. “You recognized the limitations and discussed these. The main limitiation from my point of view if the groups are not comparable by the age. However, you control it with different analyses. The other important limitation was the power of the achieved sample size. It is a great reason for continuos the study in a further step.”

Response:

We agree that these are significant limitations. It would have been more optimal to have age-matched groups, and we acknowledge that the sample size constitutes an important limitation.

Reviewer #2

Comment 18. “This study provides valuable insights into the relationship between hearing loss and emotional recognition. However, several points require attention for improvement..”

Response:

Thank you for your positive assessment.

Comment 19. “The authors should address how relying on air-conduction audiometry instead of comprehensive audiometric profiles may impact the reliability of their findings, as this is a minor deviation from the protocol”

Response:

We agree that this is a limitation that may affect the reliability of our findings. We have addressed this in the limitations subsection of the discussion, on page 46 .

Comment 20. “The reported effect size (η² = 0.02, f = 0.14) suggests the study is only powered to detect very small effects. With a sample size of N=70 participants, this raises concerns about the robustness of conclusions, especially in clinical contexts where effect sizes are typically larger (Cohen, 1988). Consequently, the findings may not generalize to the broader population of individuals with hearing loss, as significant effects in psychology and audiology often necessitate larger samples for reliable conclusions (Maxwell & Delaney, 2004).

1. Cohen J. Statistical power analysis for the behavioral sciences. 2nd ed. New Jersey, NJ: Lawrence Erlbaum Associates; 1988.

2. Maxwell SE, Delaney HD, Kelley K. Designing experiments and analyzing data: A model comparison perspective. Third edition / Scott E. Maxwell, Harold D. Delaney, and Ken Kelley. | New York,: Routledge; 2017.”

Response:

We acknowledge that the study suffers suffersfrom a lack of power to detect small effects. The initial target sample size initial target sample sizewould have allowed us to allowed us to obtain significance for small but not minimal (η² = 0.01) effects. However, due to the smaller sample size of 41 participants, there are concerns regarding the detectable effects.. A sensitivity analysis conducted later indicated that we had sufficient power to detect medium-sized effects, which enabled us to detect several of the effects of interest (in relation to our predictions), however, not all. As even small effects can be of interest, we believe future studies should aim to overcome this limitation by including larger samples, ideally more than 70 participants, to improve reliability and generalizability.

Comment 21. ”Why happiness improved specifically. Does this finding correlate with existing literature on speech perception, and could it be linked to particular frequency bands? Provide mechanistic insights or referencing previous research.”

Response:

Regarding the acoustic characteristics of happiness, we note that happiness was characterized by a particularly high pitch. A previous study (ref 9), which we now mention in the discussion (page 42 ), shows that the recognition of happiness in speech is associated with several frequency-related parameters, including a positive correlation with pitch.

Comment 22. “Why non-verbal improvements do not mirror those observed in verbal expressions. Could frequency-related parameters elucidate this difference? Discuss further”

Response:

The results of our study unfortunately do not provide reliable answers to these questions, and the limited number of recordings of nonverbal expressions do not allow us to analyze the relationship between accuracy for different emotions and acoustic differences. We describe various frequency-related characteristics of different emotions in the Results under the new subheader “Acoustics characteristics of nonverbal emotion expressions”. These indicate, for example, that many frequency-related parameters for the nonverbal expression of happiness for example are not distinct compared to other emotions (with particularly high or low values), while some frequency-related characteristics of those nonverbal emotions improved are more distinct. This does not show, however, that such characteristics were related to the improvements in their recognition. We mention a reference in the discussion on page 42 (ref 12), which shows that the recognition of the nonverbal expressions of anger and fear was correlated with pitch. As such improvement for the nonverbal expressions of fear (and the somewhat smaller improvement of anger) could be related to clearer perception of pitch.

Comment 23. “While the similar confusion patterns between groups are noteworthy, the implications of these findings (e.g., perceptual imprecision versus qualitative differences) should be discussed further, considering how altered pitch perception may influence these findings”

Response:

Thank you, we have now deepened the discussion of our findings regarding patterns of confusion further in relation to the acoustic results. See e.g. page 42, and page 44 .

Comment 24. “The repetition of results previously covered in the "Results" section dilutes clarity. Instead, the discussion should focus on how the findings align with or challenge previous similar studies”

Response:

Thank you. We reiterate results in the discussion to elaborate on our findings in relation to the predictions made in the introduction in order to to review how well the predictions were fulfilled.Additionally, we include discussions regarding how our study aligns with previous research on vocal emotion recognition and the effects of amplification in individuals with HL. We do this to provide the reader with at comprehensive understanding of our findings. Nevertheless, we have made general changes to the discussion to reduce redundance and ensure the discussion is less repetitive. We hope that the discussion will be more comprehensive, yet still accessible.

Reviewer #3

Abstract

Comment 25. “It may be beneficial to briefly define phrases such as "linear amplification" for readers unfamiliar with audiological terminology.”

Response:

Thank you, we agree that this may be beneficial. We have now added a brief definition in the abstract: “ ..,we examin

---

## [Decision Letter · Decision Letter 1]

24 Feb 2025

Dear Dr. Ekberg, 

Thank you for submitting your manuscript to PLOS ONE. After careful consideration, we feel that it has merit but does not fully meet PLOS ONE’s publication criteria as it currently stands. Therefore, we invite you to submit a revised version of the manuscript that addresses the points raised during the review process.

We look forward to receiving your revised manuscript.

Kind regards,

Hina Hadayat Ali, Ph.D

Academic Editor

PLOS ONE

Journal Requirements:

**Additional Editor Comments:**

Ensure consistency in using "identification" or "recognition." Standardize terminology e.g., "non-verbal vocalizations", correct minor errors, refine table labels, clarify descriptions, and improve phrasing for accuracy and readability.

Reviewers' comments:

Reviewer's Responses to Questions

**Comments to the Author**

1. Does the manuscript adhere to the experimental procedures and analyses described in the Registered Report Protocol?

Reviewer #3: Yes

Reviewer #5: Yes

2. If the manuscript reports exploratory analyses or experimental procedures not outlined in the original Registered Report Protocol, are these reasonable, justified and methodologically sound?

Reviewer #3: Yes

Reviewer #5: Yes

3. Are the conclusions supported by the data and do they address the research question presented in the Registered Report Protocol?

Reviewer #3: Yes

Reviewer #5: Yes

4. Have the authors made all data underlying the findings in their manuscript fully available?

Reviewer #3: Yes

Reviewer #5: Yes

5. Is the manuscript presented in an intelligible fashion and written in standard English?

*PLOS ONE*

Reviewer #3: Yes

Reviewer #5: Yes

Reviewer #3: The authors have addressed all my comments and suggestions. The manuscript is ready for the publication.

Reviewer #5: The authors have greatly improved the clarity and coherence of their manuscript. Their excellent integration of acoustic and perceptual data contributes new knowledge to the field.

Minor comments:

General comment: The authors should check if they prefer to use "identification" or "recognition" and do so consistently, as their terminology shifts between the two terms (technically two different concepts).

Abstract: "non-verbal expressions stimuli" would be better labeled as "non-verbal vocalizations", to be consistent with terminology in the introduction.

The final sentence in the "Stimuli material" section is missing a period.

Table 6:

(1) The emotions could be labeled as "reference" rather than "baseline", to be consistent with the terminology in the text.

(2) The emotion rows in the table that reproduce the reference emotion (e.g., blank "Anger" row in the first section with "Anger" as baseline) could probably be deleted for clarity, as they're not applicable.

(3) A more detailed description of the table, and a brief explanation about what the odds ratios mean (or perhaps an example), would be helpful, e.g., "Odds ratios calculated from significant coefficients in multinomial logistic regression models with each emotion as the reference emotion... Every unit increase in the Frequency component meant a 93% increase in the odds that an utterance was Happy rather than Angry" or something like that.

(4) "significand" contains a typo.

Table 8: Instead of "descriptive data", the authors could specify if this is proportion accuracy.

"Effects of hearing loss" section: It should be "18 percentage points higher", not "18% higher". Likewise for the remainder of this section.

Table 9: Based on the confusion matrices for both types of stimuli and the authors' own conclusion that "the difference between the groups regarding confusions was more quantitative than qualitative, and the tendency was for linear amplification to reduce the frequency of confusions but not the patterns", I would support having an 'X' for Prediction #5, as the prediction was that the two groups would differ in their patterns.

Conclusion: I suggest using a straightforward "effective" rather than "not ineffective".

**Do you want your identity to be public for this peer review?** For information about this choice, including consent withdrawal, please see our Privacy Policy

Reviewer #3: No

Reviewer #5: No

---

## [Author Response · Author response to Decision Letter 2]

7 Mar 2025

Comment 1: “Please review your reference list to ensure that it is complete and correct. If you have cited papers that have been retracted, please include the rationale for doing so in the manuscript text, or remove these references and replace them with relevant current references. Any changes to the reference list should be mentioned in the rebuttal letter that accompanies your revised manuscript. If you need to cite a retracted article, indicate the article’s retracted status in the References list and also include a citation and full reference for the retraction notice”

Response:

We have reviewed the reference list and made some minor corrections/changes:

1. Reference 1: We added the correct page numbering for the article.

2. Reference 6: We added a missing number to the page numbering.

3. Reference 44: We replaced the hyperlink containing the doi with the format used for doi for all other references.

4. We added the DOIs for references 46 and 47.

The references are now correct. We have not found any retracted publications among our references. We note that reference 13 received a correction in 2015 as can be seen here: https://www.pnas.org/doi/10.1073/pnas.1508604112?url_ver=Z39.88-2003&rfr_id=ori:rid:crossref.org&rfr_dat=cr_pub%20%200pubmed. However, this correction only consists of a clarification regarding methodology. It therefore has no impact on our citation of the article.

Comment 2: “Ensure consistency in using "identification" or "recognition." Standardize terminology e.g., "non-verbal vocalizations", correct minor errors, refine table labels, clarify descriptions, and improve phrasing for accuracy and readability”

Response:

Thank you for this summary of points that need addressing. We have standardized the terminology, using “recognition” and “non-verbal vocalizations” throughout the text, corrected some spelling errors and errors in phrasing, such as using “%” instead of “percentage points”, and improved the descriptions of some of the tables. Below we list the changes we have made in response to specific comments.

Comment 3: “The authors should check if they prefer to use "identification" or "recognition" and do so consistently, as their terminology shifts between the two terms (technically two different concepts)”

Response:

We have chosen to use “recognition” and “recognize” consistently throughout the text, as this is in accordance with the original phrasing in the registered report and the title of the article. Consequently, all instances of the term “identification” and “identify” have been changed. The title of the project under which the supporting data is stored at the OSF has also been changed from “identification” to “recognition”.

Comment 4: “"non-verbal expressions stimuli" would be better labeled as "non-verbal vocalizations", to be consistent with terminology in the introduction”

Response:

Thank you. We agree with this and have changed the phrasing to “non-verbal vocalizations” throughout the text.

Comment 5: “The final sentence in the "Stimuli material" section is missing a period.”

We have added a period to the final sentences in “Stimuli material”

Comment 6:

“Table 6:

1)The emotions could be labeled as "reference" rather than "baseline", to be consistent with the terminology in the text”

Response:

We have changed the label in Table 6 from “baseline” to “reference”.

Comment 7: “2) The emotion rows in the table that reproduce the reference emotion (e.g., blank "Anger" row in the first section with "Anger" as baseline) could probably be deleted for clarity, as they're not applicable”

Response:

We have deleted the blank rows that reproduce the reference emotion, as suggested.

Comment 8: “(3) A more detailed description of the table, and a brief explanation about what the odds ratios mean (or perhaps an example), would be helpful, e.g., "Odds ratios calculated from significant coefficients in multinomial logistic regression models with each emotion as the reference emotion... Every unit increase in the Frequency component meant a 93% increase in the odds that an utterance was Happy rather than Angry" or something like that”

Response:

We have rephrased the description of table 6, and to clarify further we also included a note at the end of the table exemplifying the interpretation of one OR>1 and one OR<1 (i.e. a corresponding pair of odds ratios, since an increased odds for emotion 1 vs emotion 2 is per definition a decreased odds for emotion 2 vs emotion 1, we used Anger and Happiness as example).

Comment 9: “4) "significand" contains a typo”

Response:

We have corrected the typo.

Comment 10: “Table 8: Instead of "descriptive data", the authors could specify if this is proportion accuracy”

Response:

Thank you for this suggestion, we believe it would add clarity. We have changed the text in the label of the table and added some further information for increased clarity. It now reads:

“Table 8. Proportions of accuracy for different emotions, divided by group, listening condition and type of emotion expression. Means (standard deviations) and Rosenthal’s proportion index (Π) are presented in separate columns.”

Comment 11: “Effects of hearing loss" section: It should be "18 percentage points higher", not "18% higher". Likewise for the remainder of this section”

Response:

We have changed the phrasing from “%” to “percentage points”, or “percentage point” when followed by a noun such as “difference” (“percentage-point difference”) throughout the “Effects of hearing loss” and “Effects of amplification” sections.

Comment 12: “Table 9: Based on the confusion matrices for both types of stimuli and the authors' own conclusion that "the difference between the groups regarding confusions was more quantitative than qualitative, and the tendency was for linear amplification to reduce the frequency of confusions but not the patterns", I would support having an 'X' for Prediction #5, as the prediction was that the two groups would differ in their patterns”

Response:

We agree with this argument and that an ‘X’ is more representative of what we found than a partial fulfillment. We have therefore changed the symbol to an ‘X’. As there is no longer any prediction being described as partially fulfilled, we have also removed the “check mark within parenthesis” – (✔) – symbol from the text in the table.

We have also rephrased this in the discussion, stating that “we found no support for prediction 5”.

Comment 13: “I suggest using a straightforward "effective" rather than "not ineffective"

Response:

We agree that “not ineffective” is not straight forward. We had some concern that “effective” might be too strong of a wording in relation to our results. However, within the context of the rest of the discussion, it is clear that we are not arguing that it improves accuracy for all emotions or eliminates differences from individuals with normal hearing. Therefore, we have changed the phrasing to “effective” as suggested.

---

## [Decision Letter · Decision Letter 2]

31 Mar 2025

Effects of mild-to-moderate sensorineural hearing loss and signal amplification on vocal emotion recognition in middle-aged–older individuals

PONE-D-24-26593R2

Dear Mattias Ekberg, M.S.,

We’re pleased to inform you that your manuscript has been judged scientifically suitable for publication and will be formally accepted for publication once it meets all outstanding technical requirements.

Kind regards,

Hina Hadayat Ali, Ph.D

Academic Editor

PLOS ONE

Additional Editor Comments (optional):

Reviewers' comments:

Reviewer's Responses to Questions

**Comments to the Author**

1. Does the manuscript adhere to the experimental procedures and analyses described in the Registered Report Protocol?

Reviewer #2: Yes

Reviewer #5: Yes

2. If the manuscript reports exploratory analyses or experimental procedures not outlined in the original Registered Report Protocol, are these reasonable, justified and methodologically sound?

Reviewer #2: Yes

Reviewer #5: Yes

3. Are the conclusions supported by the data and do they address the research question presented in the Registered Report Protocol?

Reviewer #2: Yes

Reviewer #5: Yes

4. Have the authors made all data underlying the findings in their manuscript fully available?

Reviewer #2: Yes

Reviewer #5: Yes

5. Is the manuscript presented in an intelligible fashion and written in standard English?

*PLOS ONE*

Reviewer #2: Yes

Reviewer #5: Yes

Reviewer #2: The authors have sufficiently addressed all previous comments. They have improved the discussion, clarified methodological concerns, and incorporated relevant literature. I have no further concerns and recommend publication.

Reviewer #5: The authors have fully addressed all my concerns, and I have no further comments on their manuscript.

**Do you want your identity to be public for this peer review?** For information about this choice, including consent withdrawal, please see our Privacy Policy

Reviewer #2: **Yes: ** Almonzer Al-Qiami

Reviewer #5: No

---

## [Editor Report · Acceptance letter]

PONE-D-24-26593R2

PLOS ONE

Dear Dr. Ekberg,

I'm pleased to inform you that your manuscript has been deemed suitable for publication in PLOS ONE. Congratulations! Your manuscript is now being handed over to our production team.

Kind regards,

on behalf of

Dr. Hina Hadayat Ali

Academic Editor

PLOS ONE